# Innovative Strategies to Combat 5-Fluorouracil Resistance in Colorectal Cancer: The Role of Phytochemicals and Extracellular Vesicles

**DOI:** 10.3390/ijms25137470

**Published:** 2024-07-08

**Authors:** Muttiah Barathan, Ahmad Khusairy Zulpa, Sook Luan Ng, Yogeswaran Lokanathan, Min Hwei Ng, Jia Xian Law

**Affiliations:** 1Centre for Tissue Engineering and Regenerative Medicine, Faculty of Medicine, Universiti Kebangsaan Malaysia, Cheras, Kuala Lumpur 56000, Malaysia; 2Department of Medical Microbiology, Faculty of Medicine, Universiti Malaya, Kuala Lumpur 50603, Malaysia; 3Department of Craniofacial Diagnostics and Biosciences, Faculty of Dentistry, Universiti Kebangsaan Malaysia, Jalan Raja Muda Abdul Aziz, Kuala Lumpur 50300, Malaysia

**Keywords:** 5-fluorouracil resistance, colorectal cancer, extracellular vesicles, drug transport, phytochemicals

## Abstract

Colorectal cancer (CRC) is a significant public health challenge, with 5-fluorouracil (5-FU) resistance being a major obstacle to effective treatment. Despite advancements, resistance to 5-FU remains formidable due to complex mechanisms such as alterations in drug transport, evasion of apoptosis, dysregulation of cell cycle dynamics, tumor microenvironment (TME) interactions, and extracellular vesicle (EV)-mediated resistance pathways. Traditional chemotherapy often results in high toxicity, highlighting the need for alternative approaches with better efficacy and safety. Phytochemicals (PCs) and EVs offer promising CRC therapeutic strategies. PCs, derived from natural sources, often exhibit lower toxicity and can target multiple pathways involved in cancer progression and drug resistance. EVs can facilitate targeted drug delivery, modulate the immune response, and interact with the TME to sensitize cancer cells to treatment. However, the potential of PCs and engineered EVs in overcoming 5-FU resistance and reshaping the immunosuppressive TME in CRC remains underexplored. Addressing this gap is crucial for identifying innovative therapies with enhanced efficacy and reduced toxicities. This review explores the multifaceted mechanisms of 5-FU resistance in CRC and evaluates the synergistic effects of combining PCs with 5-FU to improve treatment efficacy while minimizing adverse effects. Additionally, it investigates engineered EVs in overcoming 5-FU resistance by serving as drug delivery vehicles and modulating the TME. By synthesizing the current knowledge and addressing research gaps, this review enhances the academic understanding of 5-FU resistance in CRC, highlighting the potential of interdisciplinary approaches involving PCs and EVs for revolutionizing CRC therapy. Further research and clinical validation are essential for translating these findings into improved patient outcomes.

## 1. Colorectal Cancer

Normal colonic epithelial cells respond to deoxyribonucleic acid (DNA) damage by activating appropriate cellular mechanisms to repair or eliminate aberrant cells through programmed cell death pathways [1]. However, disruptions in these mechanisms can result in aberrant proliferation and the development of neoplastic precursor lesions known as polyps, which protrude into the colonic lumen from the otherwise uniform colonic mucosa [2]. Cancer stem cells (CSCs) residing within these polyps play a crucial role in initiating colorectal cancer (CRC) and tumor formation [3]. Adenomatous polyps account for the majority of colonic polyp cases, with incidence increasing with age, making CRC predominantly a disease of older individuals. On average, the transition from developing invasive adenocarcinoma from an adenomatous polyp takes up to 10 years. This slow progression, coupled with early detection and prompt treatment, has effectively reduced CRC incidence and mortality rates [4]. However, many patients at early stages are asymptomatic, leading to the majority of CRC cases being diagnosed at advanced stages, often associated with poor prognosis due to drug resistance [5]. Additionally, risk factors for CRC include being overweight, having type 2 diabetes, consuming a diet high in red and processed meats, smoking, and excessive alcohol use. Age, ethnicity, gender, family history of CRC or polyps, inflammatory bowel disease, and inherited syndromes like Lynch syndrome and familial adenomatous polyposis (FAP) also increase the risk [4]. Previous radiation therapy in the abdominal or pelvic areas is another risk factor. Identifying these factors early can help in prevention.

Despite significant advancements in oncology, approximately 90% of cancer patients with metastatic lesions develop drug resistance, severely limiting their response to chemotherapy [6]. Drug resistance significantly impacts the survival of advanced CRC patients, with a 5-year survival rate of only around 10%. Indeed, 25% of newly diagnosed CRC patients already have metastases, and 40% are predicted to develop metastases within a year following diagnosis [7]. Overcoming drug resistance remains a major challenge and the primary cause of chemotherapy failure in this field. Currently, CRC accounts for approximately 940,000 deaths annually, ranking third among cancer-related deaths worldwide. It is the second most common cancer in women and the third in men, with projected new cases reaching 2.5 million by 2035 [8]. Consequently, extensive research efforts are underway to enhance current CRC treatment by tackling drug resistance and addressing the growing prevalence of CRC.

## 2. Current Treatments of CRC

Current treatments for CRC involve multimodal approaches. These have evolved with advancements in surgical, chemotherapeutic, targeted, and immunotherapeutic approaches [9]. Surgical treatments for early-stage CRC include polypectomy, local excision, partial colectomy, and minimally invasive surgery. Advanced CRC may require colectomy, proctectomy, or pelvic exenteration for extensive cancer spread. Chemotherapy options include adjuvant, neoadjuvant, and palliative regimens like the combination of folinic acid, fluorouracil, and oxaliplatin (FOLFOX), capecitabine and oxaliplatin (CAPOX), and folinic acid, fluorouracil, and irinotecan hydrochloride (FOLFIRI). Targeted therapies focus on inhibiting vascular endothelial growth factor (VEGF) and epidermal growth factor receptor (EGFR) pathways with drugs like bevacizumab, cetuximab, and regorafenib. The addition of monoclonal antibodies, such as anti-VEGF (bevacizumab, aflibercept) and anti-EGFR (cetuximab, panitumumab), significantly benefits patients with metastatic CRC. These antibodies increase response rates and notably enhance the efficacy of FOLFOX treatment [10]. Combinations like FOLFIRI–bevacizumab and FOLFIRI–panitumumab as second-line treatments have been shown to significantly increase disease-free survival (DFS). However, the use of anti-EGFR is limited to wild-type RAS CRC only, as RAS-mutated CRC is associated with anti-EGFR resistance, poor prognosis, and inferior clinical outcomes [11]. Immunotherapy utilizes checkpoint inhibitors like pembrolizumab and nivolumab, particularly for microsatellite instability-high (MSI-H) or mismatch repair-deficient (dMMR) CRC. Radiation therapy such as external beam radiation therapy (EBRT) and brachytherapy are also employed for rectal cancer management [12]. Chimeric Antigen Receptor (CAR) T-cell Therapy response rates in solid tumors, including CRC, have been lower than in hematologic cancers. Early studies have shown response rates ranging from 10 to 30%. Combination therapies like chemoradiation and multimodal treatment integrate surgery, chemotherapy, targeted therapy, and radiation based on disease characteristics. Overall, the comprehensive treatment landscape for CRC offers a personalized approach to improving patient outcomes through a combination of innovative strategies. Among the chemotherapeutic agents, 5-fluorouracil (5-FU) remains a cornerstone, serving as the primary choice in CRC chemotherapy. 5-FU enters cells through facilitated transport, active transport, passive diffusion, and endocytosis. 5-FU is metabolically activated to form three main active metabolites: 5-fluoro-2′-deoxyuridine monophosphate (FdUMP), 5-fluorouridine triphosphate (FUTP), and 5-fluorodeoxyuridine triphosphate (FdUTP). FdUMP inhibits thymidylate synthase (TS) by forming a stable complex with the enzyme and folate, blocking thymidine synthesis and leading to DNA damage and cell death through thymine depletion. FUTP is incorporated into ribonucleic acid (RNA), disrupting normal RNA processing and function. FdUTP can be misincorporated into DNA, causing further DNA damage and cell death. These combined effects on DNA synthesis, RNA function, and direct DNA damage make 5-FU a potent antimetabolite chemotherapy agent, particularly effective against rapidly dividing cancer cells that rely heavily on nucleotide synthesis and DNA replication [13]. Figure 1 explains the 5-FU mechanism of action in CRC cells.

## 3. Application of 5-FU in CRC

Despite being established as the primary choice in CRC chemotherapy, 5-FU monotherapy exhibits a weak response rate of only around 10% in advanced CRC, leading to the adoption of combination therapy [14]. Currently, the 5-FU-based regimen FOLFOX is considered the gold standard in the first cycle of CRC chemotherapy and has been shown to benefit stage III CRC patients by improving DFS, survival after recurrence (SAR), and overall survival (OS). FOLFIRI is recognized as the superior choice in second-line treatment [15]. Understanding the role of p53 in responding to DNA damage caused by 5-FU is essential for comprehending the drug’s mechanism of action and potential mechanisms of resistance. p53, often called the “guardian of the genome”, plays a central role in the cellular response to 5-FU-induced DNA damage. When 5-FU causes DNA damage through its metabolites (primarily FdUMP and FdUTP), it triggers the activation of p53 [13]. Once activated, p53 initiates several key processes such as cell cycle arrest, DNA repair, and apoptosis. It can also induce senescence or alter metabolism in response to 5-FU stress. The p53 status in cancer cells determines their sensitivity to 5-FU, with functional p53 increasing sensitivity to apoptosis [14]. 5-FU chemotherapy also activates the p53-Fas pathway, leading to the suppression of myeloid-derived suppressor cells (MDSC) accumulation and increased cytotoxic T lymphocyte (CTL) tumor infiltration eventually enhancing antitumor immunity against CRC [15]. This is crucial for predicting 5-FU treatment outcomes and resistance in cancer therapy.

Additionally, advanced CRC is associated with acquired and intrinsic resistance to 5-FU-based regimens, necessitating higher dosages. However, high-dose usage is associated with acute and long-term side effects [16]. Acute toxicities of 5-FU include fever, mucositis, stomatitis, nausea, vomiting, and diarrhea, while cardiotoxicity may occur due to ischemia related to coronary vasospasm and direct myocardial cell toxicity. Neurotoxicity, presenting as drowsiness, dysarthria, seizure, and metabolic encephalopathy, has also been observed. In some cases, 5-FU may lead to the development of febrile neutropenia and death [17].

## 4. Resistance of 5-FU

The resistance mechanisms of 5-FU in CRC are multifaceted and intricate, involving various cellular pathways and interactions within the tumor microenvironment (TME). One prominent mechanism involves alterations in drug transport, where both increased drug efflux and decreased drug uptake contribute to reduced intracellular drug accumulation. The upregulation of ATP-binding cassette (ABC) transporters, such as ABCB1 (P-glycoprotein), ABCG2 (breast cancer resistance protein), and ABCC1 (multidrug resistance-associated protein), facilitates the efflux of 5-FU from cancer cells, diminishing its intracellular concentration and efficacy [18,19]. Conversely, alterations in solute carrier (SLC) transporters, particularly the downregulation of nucleoside transporters like human equilibrative nucleoside transporter 1 (hENT1), lead to decreased drug uptake, further limiting 5-FU’s access to cancer cells. These changes collectively reduce the bioavailability of 5-FU within tumor cells, contributing to drug resistance [20]. In addition to drug transport alterations, evasion of apoptosis plays a crucial role in 5-FU resistance. Cancer cells employ various mechanisms to suppress apoptosis, including upregulation of anti-apoptotic proteins like B-cell lymphoma 2 (Bcl-2), B-cell lymphoma-extra large (Bcl-xL), and Myeloid leukemia 1 (Mcl-1), and downregulation of pro-apoptotic proteins such as Bax and Bim [21]. Activation of signaling pathways like NF-κB/STAT3 further promotes the expression of anti-apoptotic proteins, contributing to resistance against 5-FU-induced cell death [22]. Additionally, dysregulation of the extrinsic apoptosis pathway, involving factors like Fas and Fas ligand, can also confer resistance to 5-FU by impairing caspase-mediated apoptosis. These mechanisms collectively enable cancer cells to evade the cytotoxic effects of 5-FU, promoting tumor survival and resistance [23].

Moreover, alterations in cell cycle dynamics and DNA-damage repair kinetics contribute significantly to 5-FU resistance [24]. Cancer cells may exhibit attenuated responses to 5-FU-induced DNA damage, leading to cell cycle arrest predominantly in the G1/S and S phases. Enhanced DNA repair mechanisms, particularly non-homologous end-joining (NHEJ), facilitate the efficient repair of DNA lesions induced by 5-FU, reducing the effectiveness of the drug. Dysregulation of cell cycle checkpoints mediated by kinases like ATM and ATR further promotes cell survival and resistance to apoptosis following 5-FU treatment [25]. Furthermore, the involvement of autophagy, epithelial-to-mesenchymal transition (EMT), CSCs, interactions within the TME, epigenetic alterations, and dysregulation of microRNAs (miRNAs) also contribute to 5-FU resistance in CRC [26]. These complex interplays between cancer cells, the TME, and various molecular pathways highlight the multifactorial nature of 5-FU resistance and underscore the challenges in developing effective therapeutic strategies to overcome drug resistance in CRC and the need for combination treatments that integrate chemotherapeutic agents with targeted therapies, immunotherapies, natural compounds, and innovative delivery systems such as extracellular vesicles (EV) to effectively overcome drug resistance in CRC by enhancing drug delivery and modulating the TME [27]. Figure 2 illustrates the various mechanisms that contribute to resistance against 5-FU in CRC cells.

## 5. Combination of PCs and 5-FU

PCs surpass synthetic drugs in targeting multiple pathways for holistic disease prevention and treatment with lower toxicity and fewer side effects, offering additional synergistic interactions with biological targets due to the structural diversity. They also offer various health benefits due to their unique chemical structures and biological activities [28]. Originally evolved as plant defenses, these compounds, when consumed by humans, act as antioxidants, protect against oxidative stress, modulate gene expression, and influence enzyme activity and cell signaling pathways. They also have anti-inflammatory, anti-carcinogenic, and cardioprotective effects, enhance immune function, and possess neuroprotective and antimicrobial properties. Overall, the diverse health advantages of PCs result from their ability to interact with numerous biological processes in the human body [29,30]. Furthermore, the historical use of natural PCs in traditional medicine underscores their therapeutic potential, offering a rich source of lead compounds for the development of new synthetic analogs with improved properties. PCs, such as curcumin and bitter ginseng bases, have been utilized for centuries in cancer prevention and treatment. Their utilization offers both unique advantages and challenges compared to synthetic counterparts, owing to their intricate structures and diverse scaffolds [30,31,32]. These PCs play pivotal roles in cancer immunotherapy by reshaping the immunosuppressive TME, presenting an alternative avenue to bolster immune function within this intricate milieu. Within the TME, various immune cells such as T cells, B cells, macrophages, and MDSCs potentially interact with non-cellular components, impacting therapeutic sensitivity, particularly in cancers like CRC [33,34].

Tumor-associated macrophages, typically categorized as immunosuppressive M2 or inflammatory M1 phenotypes, wield significant influence over tumor progression. M1 macrophages exhibit antitumor functions, whereas M2 macrophages foster tumor growth and migration. PCs have demonstrated the ability to modulate these functions, acting as microtubule protein stabilizers, inducing cell cycle arrest, and suppressing tumor growth. For instance, rhodopsin inhibits tumor growth by targeting specific signaling pathways involved in macrophage polarization [35,36].

Research indicates that PCs exert multifaceted effects on cancer biology, including apoptotic cell death, cell proliferation, migration, invasion, and angiogenesis, by targeting molecular pathways and disrupting oncogenic signaling cascades. Moreover, these compounds have shown efficacy in inhibiting metastasis, underscoring their potential as adjuncts to enhance drug sensitization in chemoresistant cancers and improve therapeutic outcomes, ultimately mitigating patient suffering. Ongoing investigations seek to elucidate natural product-related targets, enhance bioavailability, mitigate drug-related toxicity, and disrupt oncogenic networks, enabling the tailored selection of effective compounds for diverse tumor types [37,38,39].

Piles of preclinical studies demonstrated that oodles of PCs exerted maximum potency in killing cancer cells; however, none of them have managed to be approved clinically for CRC treatment as a single agent. Camptothecin and Taxol (paclitaxel) isolated from *Camptotheca acuminate* and *Taxus brevifolia*, respectively, have been approved by the Food and Drug Administration (FDA) and used singly to treat certain cancers [40,41]. Although irinotecan (Campto) has been approved for CRC treatment, it is used in combination with 5-FU (FOLFIRI) as a second-line treatment in patients initially treated with FOLFOX [42,43,44,45,46]. Past decades showed copious experiments that aimed to revamp the efficiency of 5-FU by combining it with PCs. Strikingly, some PCs synergistically improved 5-FU’s efficacy via bypassing 5-FU’s resistance. Thus, the combination of PCs and 5-FU was acknowledged as an important strategy to enhance CRC chemotherapy, and understanding their mechanism of action is important in order to potentially implement this strategy in a real-world setting. Table 1 shows the potential use of natural compounds to combat 5-FU-resistant CRC cells via the abovementioned mechanism.

## 6. Curcumin

Curcumin (1E,6E)-1,7-bis(4-hydroxy-3-methoxyphenyl)-1,6-heptadiene-3,5-dione) (CUR), also known as diferuloymethane, is a polyphenol extracted from the rhizomes of *Curcuma longa* (turmeric) that has been used as herbal supplementation, cosmetics, and food flavoring and coloring. It is a yellow-colored polyphenol that exerts a wide range of therapeutic benefits including antioxidant, anti-inflammatory, antimicrobial, and anticancer [47]. As one of the most explored PCs with anticancer properties, CUR has demonstrated inhibitory effects in a broad spectrum of cancers. CUR induces cell death in prostate, breast, pancreatic, bone, lung, blood, and brain cancers via apoptosis and inhibition of the NF-κB pathway [48]. In CRC cells, CUR induces apoptosis via cell cycle arrest at G2/M and G1 phases. CRC inhibition of CUR also involves multiple molecular targets including cyclooxygenase-2 (COX-2), transcription factors (NF-kB, beta-catenin), Bcl-2 family members, death receptors (DR5, Fas), caspase proteins, and reactive oxygen species (ROS) [49,50]. 

CUR synergistically raises 5-FU potency by an increment in cytotoxicity as seen in CRC cell HT-29 as well as a reduction in the COX-2 protein. COX-2 is involved in CRC tumorigenesis, and has been observed to be overexpressed in many cancers and exhibit paramount roles in cancer immunity including promoting apoptotic resistance, proliferation, angiogenesis, inflammation, invasion, and metastasis [51]. Shakibaei and colleagues demonstrated that the apoptotic induction of CUR/5-FU is via the degeneration of mitochondria and cytochrome c release as well as the upregulation of pro-apoptotic proteins caspase-8, -9, -3, PARP, and Bax; meanwhile, it is also due to the downregulation of the anti-apoptotic protein Bcl-xL and proliferative protein cyclin D1. They further reported that CUR enhanced 5-FU sensitivity by suppressing the CSCs in mismatch repair-deficient CRC cells MMR-HCT-116, HCT116/R, and HCR116 + ch3R [52,53,54]. CSCs were associated with cancer cell self-renewal, infinite proliferation, multipotency, and metastasis. CSC-related proteins were discovered to have a direct link with EMT, a process where cells fail to maintain their epithelial mannerism and affiliate with the development of an invasive metastatic phenotype [55]. Strikingly, CUR/5-FU inhibited both CSC and EMT in HCT-116R and SW480R cells via the upregulation of EMT-suppressive miRNAs and downregulation of CSC-related polycomb repressive complex subunits (BMI1, SUZ12, EZH2) [56]. 

A study of resected CRC tumors from postoperative patients uncovered a significant elevation of nuclear factor erythroid2-related factors (Nrfs) compared to normal tissues. Lee at al. discovered a significant increment in apoptosis when Nrfs RNA was altered in the SW480 cell, suggesting Nrfs protect CRC cells from apoptosis [57]. Later, the role of Nrfs was found to protect cells from oxidative damage. Remarkably, a CUR/5-FU-caused deficiency induced activated-Nrfs in resistant HCT-8R cells via Bax/Bcl-2 imbalance and led to apoptosis as well as the downregulation of P-gp and heat shock protein 27 (HSP-27) [58]. The WNT pathway is beheld as one of the driving forces of CRC initiation and tumorigenesis. This pathway is reported to be one of the necessities in the development of 5-FU resistance in CRC cells [58]. CUR/5-FU evaded resistance in HCT-116R cells via TET1-NKD2-WNT signaling where it greatly suppressed the WNT pathway and EMT [44,59]. 

In vivo, CUR chemo-sensitized 5-FU via the reduction in tumor size in an HCT-116R-xenograft mice model [60]. Furthermore, a phase I clinical trial of CUR plus FOLFOX boosted the antiproliferative and apoptotic effects of patient-derived explants while alleviating CSCs markers [61]. In a phase IIa trial, oral CUR plus FOLFOX was reported to be safe and tolerable in metastatic CRC chemotherapy [62]. 

## 7. Resveratrol

Resveratrol (3,5,4′-trihydroxy-trans-stilbene) (RES) is a phytoalexin, stilbenoid, and natural polyphenol produced by grapes, blueberries, mulberries, and peanuts in response to injury or pathogenic attack [63]. RES can be found in more than 70 plants and is prominent for its huge antioxidant and chemopreventive capacity [64]. The chemopreventive effects of RES were validated in several carcinogenesis mouse models including dimethylbenzanthracene (DMBA)-initiated and 12-O-tetradecanoyl-13-acetate (TPA)-induced skin cancer, N-methylnitrosourea (MNU)-induced breast cancer, diethylnitrosamine plus 2-acetylaminofluorene-induced liver cancer, prostatic adenocarcinoma in the transgenic adenocarcinoma mouse prostate (TRAMP), and benzo[a]pyrene (BP)-induced lung cancer [65,66]. As an anticancer agent, RES induces apoptosis in many cancer cells including breast, ovarian, cervical cancers, and CRC. RES exploited CRC DLD-1 and HCT-15 cells via the AKT/STAT3 signaling pathway with the formidable induction of apoptotic-mediated cell death at phase G1 cell cycle arrest [67,68]. Moreover, RES generates the declination of cyclin D1, E2 and BCL-2, while it elevates the BCL-2x and p53 proteins [69]. 

In attempt to discover combination of RES and 5-FU activity, Chan et al. found that RES/5-FU induces apoptosis via caspase-6 activation in both wild-type and knockout p53 HCT-116 cells [70]. This significant work demonstrated that RES is able to induce apoptosis despite the p53 mutation that is implicated with the apoptotic resistance and poor clinical outcome observed in ~50% of sporadic CRC [71]. Furthermore, RES/5-FU induces cell cycle arrest at S phase with dominant DNA damages via activation of the mitogen-activated protein kinase (MAPK) pathway, an important signaling pathway in the regulation of cancer cells’ proliferation by upregulation of the p-JNK and p-p38 proteins [72]. Additionally, the effect of RES/5-FU on the STAT3 pathway has been validated in HT-29, SW620, DLD1, and HCT-116 cells. STAT3 was implicated with angiogenesis, cell migration, and drug resistance in cancers by promoting mitochondrial transcription and oxidative respiration that protect cancer cells against oxidative damage. It is typically overactive in ~70% of solid tumors [73]. Notably, RES/5-FU sensitizes CRC cells to oxidative stress with a subsequent surge in apoptotic events via the deletion of STAT3-related proteins [74]. 

Moreover, RES/5-FU directly inhibits EMT in HCT-116R cells via the upregulation of E-cadherin factor, and the loss of this entity leaves cells prone to the deprivation of adherence, leading them to become motile and invasive with an acquired mesenchymal nature that is ratified as a hallmark of EMT. RES/5-FU also repressed the EMT-linked pathway NF-kB through the inhibition of TNF-β-mediated activation. NF-kB pathway activation promotes CRC cells’ proliferation, suppresses apoptosis, induces angiogenesis, and enhances EMT-mediated survival and metastasis. It was also reported that TNF-β activation favors upregulation of CSCs phenotypes (CD133, CD44, and ALDH1) of resistant CRC cells. Notably, RES/5-FU abolishes NF-kB-related genes and downregulates TNF-β, suggesting it also suppresses CSC survival [16]. Moreover, the anticancer properties of RES/5-FU have been validated in vivo whereby treatment on HCT-116R xenograft mice showed potent tumor shrinkage [75]. In N-methylnitrosourea-induced CRC rats, RES/5-FU suppressed NF-kB activation while yielding a protective effect on the colonic tissue as evidenced by an improved intact surface epithelium, normal colon cells, and reduction in inflammation [76].

## 8. Epigallocatechin Gallate

Green tea, derived from the buds of *Camellia sinensis* plants, is a cherished beverage in Asia, particularly in China and Japan. Rich in polyphenols, with epigallocatechin gallate (EGCG) being its most potent antioxidant, green tea has gained recognition for its chemopreventive properties [77]. Studies have shown that EGCG effectively induces apoptosis in various cancer cell types, including those associated with nasopharyngeal, breast, prostate, liver, bladder, and ovarian cancers, primarily through the modulation of Bcl-2 family proteins and the generation of reactive oxygen species (ROS) [78]. In CRC, EGCG treatment has been linked to the upregulation of caspase-3/7 and PKR-like endoplasmic reticulum kinase (PERK), pivotal players in endoplasmic reticulum (ER) stress-mediated apoptosis [79].

Furthermore, research has demonstrated the synergistic enhancement of cytotoxicity when EGCG is combined with 5-FU. This combination therapy induces apoptosis in CRC cells via caspase-3 and PARP activation while also affecting key signaling pathways such as PI3K/Akt, known for regulating cancer cell survival and proliferation [80]. Notably, EGCG/5-FU treatment has been found to downregulate the expression of multidrug resistance protein 1 (MDR1/ABCB1), a gene responsible for drug efflux and multidrug resistance in cancer cells. This effect occurs through the NF-kB/miR-155-5p pathway, potentially overcoming resistance mechanisms observed with 5-FU alone [81].

In animal models, including xenograft mice, the combination of EGCG and 5-FU has shown significant suppression of tumor growth compared to individual treatments, suggesting its potential as an effective therapeutic approach for CRC [82]. Additionally, in 3D spheroid models, EGCG/5-FU treatment has been associated with the downregulation of CSC markers and the upregulation of self-renewal suppressive miRNAs, further indicating its ability to target and reverse resistance, particularly through the inhibition of CSCs [83].

## 9. Genistein

Genistein (4′,5,7-trihydroxyflavone), a phytoestrogen belonging to isoflavones abundant in soy products, possesses diverse therapeutic capacities, including addressing menopause-related conditions like cardiovascular issues and osteoporosis, as well as exhibiting chemopreventive and anticancer properties [84]. First isolated from *Genista tinctoria* L. and found in various plants including *Trifolium* species and *Glycine max* L., genistein demonstrates cytotoxic effects against several non-hormone-dependent cancers such as colorectal, gastric, pancreas, and lung cancers [85].

Regular consumption of soy-based products has been associated with a reduced incidence of CRC, believed to be mediated partly by genistein [86]. In CRC cells like HT-29, genistein inhibits cell migration and reverses EMT by modulating the expression of E-cadherin and N-cadherin. It also affects key signaling pathways such as Notch-1, NF-kB, and PI3K/Akt, leading to cell cycle arrest and apoptosis through alterations in protein levels of Bax, Bcl-2, and caspase-3 [87,88].

Moreover, genistein synergistically enhances the cytotoxicity of 5-FU in HT-29 cells by promoting apoptosis via increased cleavage of pro-apoptotic PARP and expression of p53 [89]. This combination therapy uniquely downregulates the survival signal glucose transporter 1 (Glut-1), crucial for cancer cell glucose metabolism, while upregulating p21, a cell cycle inhibitor protein often dysregulated in cancer cells [90]. Additionally, genistein/5-FU treatment leads to ROS accumulation, activating the MAPK pathway, and inhibits COX-2 production in CRC cells, suggesting a potential role in inflammatory regulation [91].

Furthermore, genistein mediates an anti-inflammatory response by reducing cytokine levels and improving colonic permeability and barrier function through TLR4/NF-kB signaling in colonic injury mice. This is particularly relevant as 5-FU has been implicated in causing intestinal mucositis via pro-inflammatory mechanisms in vivo [92]. Therefore, the combinational approach of genistein/5-FU could offer benefits for patients with colonic injury while potentially mitigating drug resistance issues.

## 10. Geraniol

Geraniol ((2E)-3,7-Dimethyl-2,6-octadien-1-ol), a monoterpenic alcohol found in various essential oils, is rapidly isolated from numerous plants including *Pelargonium graveolens*, *Cymbopogon martinii* var motia, *Cymbopogon winterianus* Jowitt syn, *Cymbopogon nardus* L., *Cymbopogon winterianus*, and *Cymbopogon jwarancusa* [93]. Extensive evidence supports its chemopreventive and anticancer activities across various human cancers, including breast, lung, colon, prostate, and pancreatic cancers, primarily through the elevation of pro-apoptotic proteins [94].

In CRC, geraniol induces apoptosis by upregulating Bax, downregulating Bcl-2, causing DNA damage, and inducing G2/M cell cycle arrest. Moreover, geraniol enhances the effectiveness of commercial anticancer drugs such as docetaxel (DTX) and 5-FU in prostate and colon cancers, respectively. Combination therapy with geraniol and DTX suppresses prostate cancer growth both in vitro and in tumor xenograft mice. Similarly, the combination of geraniol and 5-FU significantly enhances the inhibition of CRC cells compared to 5-FU alone, attributed to geraniol’s ability to increase the intracellular uptake of 5-FU, thereby boosting its bioavailability [95,96].

Furthermore, geraniol/5-FU combination therapy significantly reduces the levels of crucial enzymes for DNA synthesis, TS, and thymidine kinase (TK), thereby enhancing the DNA damaging effects of 5-FU mediated via FdUMP. In animal models, geraniol/5-FU treatment demonstrates impressive reductions in tumor volume, with a significant decrease observed in patient-derived tumor xenograft mice, validating the sensitization effect of geraniol in enhancing the efficacy of 5-FU in vivo [97].

## 11. Thymoquinone

*Nigella sativa*, commonly known as black cumin, holds a significant place in traditional remedies and is recognized as one of the most valuable nutrient-dense herbs globally. Thymoquinone (TQ), a bioactive compound abundantly found in *Nigella sativa*, offers powerful therapeutic benefits against a wide array of diseases, including cardiovascular complications, diabetes, asthma, kidney disease, and cancer [98]. TQ exhibits cytotoxic effects on various cancers, including those affecting blood, bladder, breast, cervical, colorectal, gastric, brain, liver, oral, lung, prostate, and pancreatic tissues, primarily through alterations in DNA structures [99].

In CRC, TQ’s chemopreventive potential has been demonstrated in vivo, where it mitigates oxidative stress in erythrocytes during colon carcinogenesis induced by 1,2-dimethylhydrazine (DMH) in male Wistar rats [100]. TQ induces apoptosis in CRC cells such as HCT-116 by modulating the expression of pro-apoptotic and anti-apoptotic proteins, including Bax, Bcl-2, Bcl-xl, caspases, and PARP. Additionally, TQ inhibits STAT3 signaling, thereby affecting downstream targets such as survivin, c-Myc, cyclin-D1, and cyclin-D2 while promoting the expression of anti-cell cycle proteins p27 and p21 [101,102].

Moreover, TQ induces apoptosis and DNA damage, and suppresses CSC markers in CRC cells, leading to the inhibition of migration and invasion. Combination therapy with 5-FU enhances TQ’s anticancer efficacy by targeting CSCs and downregulating the WNT/beta-catenin and PI3K/Akt pathways [103]. In vivo studies corroborate these findings, showing the inhibition of tumor growth in CRC xenograft mice treated with TQ alone or in combination with 5-FU, accompanied by alterations in signaling pathways associated with cell adhesion and survival [104]. These findings underscore TQ’s potential as a therapeutic agent against CRC, particularly through its targeting of CSCs, and warrant further clinical investigation to validate its efficacy in human patients. Figure 3 highlights the potential PCs that can be used in combination with 5-FU to enhance its efficacy and overcome resistance in CRC treatment.

## 12. EVs as Communication Mediators

EVs have undergone a remarkable journey of discovery and exploration since their initial identification as cellular debris, with key milestones shaping our understanding and applications. Emerging in the 1980s and 1990s, their significance in intercellular communication gradually unfolded, revealing diverse types such as exosomes, microvesicles, and apoptotic bodies, each with distinct biogenesis pathways and functions. 

Small EVs, or exosomes, are nano-sized vesicles originating from intraluminal vesicles (ILVs) within multivesicular bodies (MVBs). The formation of ILVs within MVBs can occur through three main mechanisms. The first involves endosomal sorting complexes required for transport (ESCRT) complex members, which select and segregate specific cargoes into microdomains on the endosomal membrane. The second pathway, independent of ESCRT, relies on proteins like ALIX and transmembrane proteins to recruit cargoes and facilitate vesicle budding [105]. The third mechanism involves membrane lipid microdomains or lipid rafts, with ceramide playing a crucial role in vesicle membrane bending. Following their formation, MVBs can either fuse with lysosomes for degradation or fuse with the plasma membrane to release exosomes [106].

Medium/large EVs, also known as microvesicles, range in size from 50 to 1000 nm and are released from the cell surface by blebbing from the plasma membrane. Their biogenesis invoves various partners, including small GTPases, Ras-related proteins, and phospholipases, ultimately leading to the recruitment of extracellular signal-regulated kinase (ERK) and phosphorylation of myosin light-chain kinase (MLCK) to induce plasma membrane invagination and EV release [107].

Apoptotic bodies (APOBs), the third subtype of EVs, vary in size and are produced during programmed cell death (apoptosis). APOBs exhibit specific sorting mechanisms for organelles, RNA, and DNA fragments, distinguishing them from other EV subtypes. Despite their association with cell death, APOBs are increasingly recognized for their involvement in various biological processes [108,109].

The PCs within EVs, including proteins like tetraspanins and RNA molecules, are integral to their structure, cargo loading, and function, enhancing their therapeutic potential. Understanding the biogenesis pathways of EVs, from the endosomal pathway for exosomes to direct budding from the plasma membrane for microvesicles, is crucial for both basic research and therapeutic development. Collectively, these milestones have provided a comprehensive background of EVs, illuminating their PCs and potential applications across diverse fields.

EVs offer a promising strategy for targeted drug delivery in cancer treatment, especially for delivering 5-FU and natural compounds. These small membrane-bound particles, released by cells, facilitate the transfer of bioactive molecules between cells. Engineered EVs in cancer therapy can directly transport therapeutic agents to cancer cells, thereby enhancing efficacy and minimizing toxicity. They effectively protect cargo, improve drug uptake, and overcome resistance mechanisms while being less immunogenic than synthetic carriers. By delivering 5-FU and PCs to cancer cells, EVs enable reduced drug dosages, fewer side effects, and improved treatment outcomes. This approach synergizes the anticancer effects of drugs with targeted EV delivery for more effective cancer treatment [27]. Moreover, exosomes loaded with natural PCs have shown significant promise in cancer therapy. Milk-derived exosomes carrying curcumin and anthocyanidins have demonstrated enhanced antiproliferative and anti-inflammatory effects in models of lung cancer [110]. Similarly, exosomes loaded with withaferin A and paclitaxel exhibit substantial antitumor effects, with withaferin A-loaded exosomes demonstrating twice the tumor inhibitory effect compared to free drug administration [111]. These findings underscore the potential of phytochemical-loaded exosomes to significantly enhance cancer treatment outcomes.

## 13. EV-Mediated CRC Progression

The interaction between EVs and cell membranes triggers EV uptake through various routes, including direct fusion, endocytosis, macropinocytosis, and phagocytosis. The uptake process depends on factors such as EV subtype, composition, recipient cell membrane characteristics, and extracellular conditions. EV exchange plays a crucial role in intercellular communication, influencing physiological regulation across different tissues. Additionally, dysfunctions and diseases, including cancer progression, can be fueled by altered EV interactions. EVs have been implicated in various hallmarks of cancer progression, including proliferation, evasion of growth suppressors, resistance to cell death, induction of angiogenesis, and metastasis support, highlighting their significance in disease pathogenesis [112]. CRC-derived exosomes harbor a diverse array of biomolecules, including proteins (e.g., TGFβ, VEGFA, αvβ5 integrin, PAD4, WNT4), RNA (e.g., miR-25-3p, miR-21, miR-27a, miR-10b, miR-2149-5p, miR-6737-5p, miR-6819-5p), and DNA. These exosomes play pivotal roles in modulating various aspects of tumor progression and metastasis [113]. 

For instance, CRC cells can induce the formation of cancer-associated fibroblasts (CAFs) by stimulating macrophages to secrete TGF-beta-containing exosomes [114]. CAFs are fibroblasts that have been activated within the TME, and they are recognized as the most prevalent type of tumor mesenchymal cells. Their abundance and activity within the TME play crucial roles in tumor progression, angiogenesis, immune evasion, and other aspects of cancer development and metastasis [115]. Furthermore, CAFs promote EMT and CSC-like phenotypes in colon cancer cells, contributing to chemoresistance. This study found that Circ_0067557, circular RNA, is highly expressed in CAF-derived exosomes and promotes CRC progression and chemoresistance by targeting Lin28A and Lin28B. These Lins are primarily known for inhibiting the let-7 family of miRNAs, which are essential regulators of cell proliferation, differentiation, and development [116]. 

Similarly, another study has mentioned that exosomes derived from colon cancer cells promote EMT by inducing changes in protein expression such as downregulation of E-cadherin and upregulation of N-cadherin and β-catenin. In addition, these exosomes interact with the TME, including CAFs, and contribute to premetastatic niche formation [117]; meanwhile, a study found that miR-24-3p expression is increased in colon cancer tissues and cells. Overexpression of miR-24-3p promotes cell viability and colony formation, and inhibits apoptosis in colon cancer cells treated with methotrexate (MTX). MiR-24-3p is enriched in CAFs-derived exosomes and can be transferred to colon cancer cells. Exosomal miR-24-3p from CAFs promotes tumor growth and resistance to MTX by targeting the CDX2/HPEH regulatory axis [118]. CAFs release exosomes containing a higher level of miR-17-5p compared to normal fibroblasts (NFs). This exosomal miR-17-5p can be transferred from CAFs to CRC cells, promoting their metastatic potential. miR-17-5p directly suppresses the gene RUNX3, which plays a role in regulating cell behavior leading to tumor progression [119]. 

In hypoxic condition, CRC cell exosomes carry Wnt4 mRNA molecules and interact with endothelial cells (cells lining blood vessels), and this activates β-catenin signaling in endothelial cells. Activated β-catenin signaling promotes the proliferation and migration of endothelial cells, which are key steps in angiogenesis [120]. In another study, B7-H3, an immune checkpoint molecule, was shown to be overexpressed in both cancer cells and vascular endothelial cells (VECs) in CRC tissues. B7-H3 in exosomes activated the AKT1/mTOR/VEGFA signaling pathway in VECs, leading to increased migration, invasion, and angiogenesis in CRC, and this eventually led to lung metastasis [121]. Additionally, specific integrins within CRC-derived exosomes have been associated with liver or lung metastases. Exosomal PAD4 affects the citrullination of the extracellular matrix (ECM), promoting colon cancer EMT and liver metastasis [122]. 

CRC cell-derived exosomal HSPC111 converts fibroblasts into CAFs within the liver premetastatic niche, promoting CRC liver metastasis (CRLM). Furthermore, elevated expression of HSPC111 in patient serum exosomes and primary CRC tissues correlated positively with liver metastasis. Exosomal HSPC111 upregulated acetyl-CoA levels in CAFs, influencing lipid metabolism through ACLY phosphorylation, a process associated with cancer growth and metastasis. The study revealed that exosomal HSPC111 treatment specifically altered H3K27 acetylation in CAFs, highlighting its role in promoting CRLM. Notably, exosomal HSPC111 derived from CRC cells induced a range of cancer-promoting factors in CAFs, including CXCL5, TGF-β, MMP2, and SERPINE1, enhancing their tumor-promoting effects [123].

Exosomal miRNAs also play critical roles in CRC progression. CAF-derived exosomal miR-135b-5p is upregulated in both CAF exosomes and CRC tissues/cells. This promotes CRC growth and angiogenesis by inhibiting thioredoxin-interacting protein (TXNIP) expression. TXNIP was found to be downregulated in CRC tissues/cells. This research also mentioned that overexpression of TXNIP weakens the pro-cancer effects of exosomal miR-135b-5p [124]. MiR-21 promotes CRC cell proliferation, invasion, and therapy resistance by targeting miR-21 functions by suppressing the expression of tumor suppressor genes such as programmed cell death 4 (PDCD4), tropomyosin 1 (TPM1), and phosphatase and tensin homolog (PTEN) genes, which normally act to restrain cancer progression [125]. miR-2149-5p, miR-6737-5p, and miR-6819-5p inhibit TP53 expression in fibroblasts to promote tumor proliferation [126]. CRC releases exosomal miR-10b that suppresses the PIK3CA gene, which in turn inhibits the PI3K/Akt/mTOR pathway. This pathway is crucial for normal cell growth and survival, while activating the production of TGF-β and α-SMA. These molecules are associated with the transformation of fibroblasts into CAFs [127]. MiR-224-5p is enriched in CAFs-derived EVs and can promote CRC cell proliferation, migration, and invasion, and inhibit apoptosis by targeting SLC4A4 expression [128]. Exosomes derived from CAFs enhance miR-181b-3p expression in CRC cells, promoting cell proliferation and migration while decreasing apoptosis by targeting SNX2 [129]. Hypoxia induces the secretion of circEIF3K from CAFs into CRC cells via exosomes, promoting malignant growth and metastasis by regulating the miR-214/PD-L1 axis [130]. Another study has mentioned that CAFs release exosomes that contain miR-625-3p. CRC cells take up these exosomes, and the miR-625-3p within them promotes invasion, migration, and chemotherapeutic resistance [131]. 

In a recent study, researchers found that conditioned media (CM) and exosomes derived from chemoresistant HCT-15/FU cells promoted angiogenesis more effectively than those from HCT-15 cells. The study identified that HCT-15/FU exosomes contained high levels of growth differentiation factor 15 (GDF15), which enhanced angiogenesis by inhibiting the Smad signaling pathway and increasing the expression of periostin (POSTN) [132]. Meanwhile, miR-181d-5p, enriched in CAFs, is associated with sensitivity to 5-FU. CAF-derived exosomes inhibit 5-FU sensitivity in CRC cells via the METTL3/miR-181d-5p/NCALD axis. CricN4BP2L2, enriched in CAFs exosomes, promotes oxaliplatin resistance and stemness in colon cancer cells while inhibiting apoptosis. It regulates the PI3K/AKT/mTOR pathway by binding to EIF4A3, thus contributing to drug resistance [133].

Exosomes released from oxaliplatin-resistant colon cancer cells (LS174T/R) can spread resistance to sensitive cells (LS174T/S) due to a high level Nrf2 protein expression, which plays a role in drug detoxification. Sensitive cells that take up these exosomes also increase their Nrf2 levels. This confirms the role of exosomes in promoting drug resistance in CRC [134]. The miR-92a-3p from stromal cell exosomes induces CRC cell EMT and inhibits mitochondrial apoptosis through activation of the Wnt/β-catenin pathway and suppression of the FBXW7 and MOAP1 genes. High levels of exosomal miR-92a-3p in serum are associated with metastasis and chemoresistance in CRC patients [135].

Another study employed proteome profiling analysis followed by validation studies, revealing for the first time the enrichment of p-STAT3 in exosomes and its relevance to 5-FU resistance in CRC. The findings underscore the importance of conducting large-scale mass spectrum-based analyses to screen potential chemoresistant proteins in exosomes. Phosphorylated STAT3 (p-STAT3) is known to be activated in various cancers, including CRC, and has been associated with patient survival and response to chemotherapy. While few studies have explored exosomal p-STAT3 in CRC, the mechanisms regulating its enrichment in exosomes remain poorly understood. The study’s findings suggest that developing inhibitors targeting p-STAT3 could be an effective strategy to mitigate exosome-mediated chemotherapy resistance in CRC. Moreover, exosomal p-STAT3 could potentially serve as a biomarker for monitoring 5-FU resistance in CRC patients during treatment [136].

In another study, exosomes isolated from *Fusobacterium nucleatum* (Fn) were used to infect CRC cells, and it was found that Fn infection transported miR-1246/92b-3p/27a-3p and CXCL16/RhoA/IL-8 into non-infected cells, thereby enhancing the cell migration ability in vitro and promoting tumor metastasis in vivo. Furthermore, circulating levels of exosomal miR-1246/92b-3p/27a-3p and CXCL16 were closely correlated with Fn abundance and tumor stage in CRC patients. This study suggests that Fn infection stimulates tumor cells to produce exosomes enriched with miR-1246/92b-3p/27a-3p and CXCL16/RhoA/IL-8, which are then delivered to uninfected cells, promoting pro-metastatic behaviors [137]. Table 2 summarizes the CRC-promoting effects of EVs.

## 14. EV-Mediated Therapy for CRC

The effective delivery of drugs into CRC cells is crucial for successful treatment, especially considering the complexities of the TME. While synthetic nanomaterial-based drug delivery systems have made significant progress, they still have limitations. Natural carriers, particularly EVs, have emerged as promising candidates due to their biocompatibility, evasion of clearance mechanisms, and homing abilities [138].

The isolation and purification of EVs are critical for their utilization as drug delivery systems. Various methods, such as centrifugation, affinity capture, gel-permeation chromatography, membrane filtration, precipitation, and microfluidic devices, are employed for EV separation [139]. While each method has its advantages and limitations, a combination of techniques is often necessary to isolate EVs effectively. Different sources of EVs, including native EVs, CRC cell-derived EVs (CEXs), bacterial outer membrane vesicles (OMVs), plant EVs (PEVs), and milk-derived EVs (MEVs), have been explored for drug delivery in CRC. CEXs, in particular, offer CRC-specific targeting potential due to their similarity to CRC cell membranes [140]. Moreover, EVs derived from various sources have shown promising results in delivering therapeutic cargo, including drugs and nucleic acids, to CRC cells [141]. EVs offer several advantages, such as protecting cargo from degradation, enhancing bioavailability, and enabling targeted delivery to tumor sites. Moreover, EVs can be engineered to carry specific molecules, such as small RNAs or proteins, that sensitize CRC cells to 5-FU [140], potentially reversing drug resistance mechanisms. Preclinical studies have shown encouraging results, demonstrating the ability of EVs to overcome resistance and improve therapeutic outcomes in CRC models [141]. Moving forward, optimizing EV-based delivery systems and validating their efficacy in clinical settings are crucial steps toward developing effective treatments for 5-FU-resistant CRC patients.

Engineered EVs, obtained through genetic or chemical modifications, improve their targeting abilities and cargo-loading efficiency, which eventually enhances the anticancer properties [142]. Despite their potential, challenges remain in ensuring uniform expression of target proteins and maintaining EV structural integrity. Bionic EVs, such as artificially synthesized EVs (ASEVs), nanovesicles (NVs), and hybrid vesicles (HVs), offer advantages in terms of large-scale production and cargo delivery [143]. Studies have shown that exosomes engineered to overexpress tumor-suppressive miRNAs (e.g., miR-379, miR-16-5p) or lacking oncogenic miRNAs (e.g., miR-424) exhibit potent inhibitory effects on CRC cell proliferation, migration, and invasion both in vitro and in vivo [144]. Engineered exosomes offer a promising approach for developing targeted and personalized therapies for CRC. However, further research is needed to fully understand their biogenesis and regulatory mechanisms [145,146].

EVs also hold promise as biomarkers for CRC prediction, diagnosis, and prognosis. Studies have identified EV-associated molecules, such as miRNAs and proteins, that can differentiate CRC from healthy tissue and predict patient outcomes. Detection methods, including transmission electron microscopy, nanoparticle tracking analysis, Western blotting, ELISA, and high-sensitivity flow cytometry, are used to characterize EV biomarkers. Efforts are ongoing to develop more accurate and cost-effective technologies for EV detection in CRC [147]. For instance, decreased plasma levels of EV-miR-193a-5p were found in CRC patients, with potential diagnostic value (area under curve, AUC, of 0.740 and 0.759 in distinguishing CRC from precancerous adenomas and non-cancerous controls, respectively). Moreover, exosomal CircLPAR1 in plasma showed improved diagnostic performance (AUC of 0.875) for CRC, correlating with overall survival. Additionally, serum EV-associated miRNA-21 and miRNA-92a levels were superior in diagnosing metastatic CRC compared to a carcinoembryonic antigen, with higher miRNA-222 levels predicting lower overall survival [148]. 

Plant-derived exosome-like nanoparticles (PENs) have garnered considerable attention as potential nanocarriers for delivering therapeutic agents, including CRISPR/Cas9 components targeting long non-coding RNAs (lncRNAs) associated with colon cancer development and progression [149]. PENs offer advantages such as biocompatibility, low immunogenicity, and the ability to traverse biological barriers, making them attractive candidates for oral delivery of CRISPR/Cas9 therapeutics to the lower digestive tract. Ginger-derived nanoparticles (GDNVs) have been extensively investigated as carriers for therapeutic agents, particularly in the context of colon cancer. Studies have demonstrated that GDNVs loaded with chemotherapeutic drugs, such as adriamycin (Dox), effectively inhibit colon cancer cell proliferation and induce apoptosis [150]. Moreover, modification of GDNVs with targeting ligands, such as folic acid, enhances their specificity for colon tumor sites, leading to improved therapeutic outcomes. Additionally, GDNVs have been utilized for targeted delivery of small interfering RNAs (siRNAs) to colonic tissues, resulting in decreased expression of specific target genes associated with cancer progression [151]. Grape-derived exosome-like nanoparticles (PDENs) have been explored for their ability to target colon tumor sites in vivo, leading to enhanced chemotherapeutic inhibition of tumor growth [152]. These nanoparticles have also been investigated in inflammation-driven tumor models, where they demonstrated improved targeting to inflammatory tissues and enhanced therapeutic effects. When grapefruit nanovesicles (GNV) are combined with folic acid (FA), these carriers exhibit enhanced tumor-targeting capabilities. Additionally, encapsulating the antitumor drug paclitaxel (PTX) on GNV-FA allows for a more specific treatment of colon cancer [153]. Furthermore, co-delivery of GNV-FA and the chemotherapeutic drug PTX has shown enhanced specificity and efficacy against colon cancer tumors in xenograft mouse models, including CT26 colon cancer and human SW620 colon cancer SCID mouse models [154]. This approach resulted in a reduction in tumor size and extended the survival of tumor-bearing mice. A study demonstrated the effects of exosomes derived from CT26 tumor cells on tumor growth and immune modulation in a mouse model of colon cancer. The results showed a significant reduction in CT26 tumor size in mice treated with tumor-derived exosomes compared to the control group without exosomal treatment. Additionally, the mean weight of mice treated with exosomes was significantly higher than that of the control group. These findings are consistent with previous research, such as studies involving exosomes derived from human embryonic kidney cells loaded with hepatocyte growth factor siRNA, which inhibited tumor growth and angiogenesis in gastric cancer models [155].

Similarly, grapefruit-derived EVs (GfEVs) have been identified as efficient carriers for delivering exogenously loaded proteins, such as Alexa fluor-tagged bovine serum albumin (BSA) and heat shock protein 70 (HSP 70), into both peripheral blood mononuclear cells and colon cancer cells. These vesicles have demonstrated safety and efficacy as carriers for exogenous proteins compared to proteins delivered without EVs [156]. Lemon-derived exosome-like nanoparticles have shown inhibitory effects on the growth of colon cancer tumors in vivo, suggesting their potential as a therapeutic intervention. Clinical trials involving exosomes for colon cancer are ongoing, albeit still in the early stages. These trials primarily focus on using exosomes as drug delivery systems or therapeutic agents [157].

Some clinical trials aim to encapsulate therapeutic agents, such as curcumin, into plant-derived exosomes for the treatment of CRC. For instance, grape-derived PDENs loaded with curcumin are being investigated for their potential therapeutic benefits in colon cancer patients (NCT01294072) and (NCT04879810) [158,159]. 

Moreover, exosomes from non-CRC origins can also remodel CRC cells. Exosomes derived from BMSCs and ascites have demonstrated protective effects against colitis, wound healing in the intestine, and regeneration of epithelial cells, suggesting their potential for CRC and inflammatory bowel disease (IBD) therapy [160]. Ascites-derived exosomes have been explored for immunotherapy of CRC in clinical trials, further highlighting the therapeutic potential of exosomes in CRC treatment [161]. ANGPTL1-containing exosomes downregulate MMP9 levels in Kupffer cells, inhibiting the JAK2-STAT3 signaling pathway and attenuating CRC liver metastasis [162]. Exosomal DNA from CRC cells can modulate tumor-associated immunity [163]. Nucleic acid-rich exosomes induce antitumor immunity by stimulating the TLR-IFN pathway, while DNA-containing exosomes trigger dendritic cell activation and subsequent CD8+ T cell activation via the CGAS-STING pathway [164]. This indicates that CRC cell-derived EVs can be used as a cancer vaccine to induce an antitumor immune response. The antitumor effects of BMSC-derived exosomes involve the inhibition of CRC cell proliferation, migration, and invasion, as well as the induction of apoptosis [165]. These effects are mediated through the downregulation of specific genes such as integrin α2 (ITGA2). Table 3 displays a summary of the research on EVs and their potential in CRC treatment.

## 15. Potential Combination of 5-FU and EV

The combination of 5-FU with EVs aims to enhance the therapeutic efficacy of 5-FU while reducing systemic toxicity. EVs can be engineered to encapsulate or load 5-FU, protecting it from degradation and improving its delivery to tumor cells. This approach improves the pharmacokinetics and biodistribution of 5-FU, potentially increasing its accumulation in tumor tissues and reducing off-target effects. Moreover, modifying EVs with targeting ligands enhances their selective uptake by cancer cells, further improving drug delivery specificity. Natural PCs, known for their anticancer properties, can be co-loaded with 5-FU into EVs. These PCs, such as those derived from ginger or grapefruit, have shown promise in preclinical studies for enhancing cytotoxicity against cancer cells and overcoming drug resistance.

One potential limitation of combining 5-FU with EVs lies in the complexity of the TME and the heterogeneity of cancer cells. Tumors can evolve diverse mechanisms of resistance to chemotherapy, making it challenging to develop universal therapeutic strategies. Additionally, the isolation and purification of EVs for clinical applications present technical challenges, including the standardization of isolation methods, scalability of production, and maintaining EV integrity and cargo stability. Moreover, while PCs and engineered EVs show promise in preclinical studies, their translation to clinical practice requires rigorous validation, optimization, and safety assessment in human trials. Furthermore, the identification and validation of EV-associated biomarkers for clinical use necessitate large-scale studies and validation in diverse patient cohorts. Additionally, the regulatory approval process for novel therapies and diagnostics involves stringent criteria and lengthy timelines, which may impede the timely translation of research findings into clinical applications. Addressing these limitations will require interdisciplinary collaboration, technological innovation, and robust clinical validation to realize the full potential of these approaches in improving outcomes for patients with CRC. Figure 4 illustrates a potential combination therapy involving 5-FU and EVs.

## 16. Potential Challenges

Complex Interactions: The relationships among natural compounds, chemotherapy drugs, and EV delivery systems are truly intricate and may be unpredictable. These interactions have a substantial impact on the way combined therapies work in the body [166]. Possible outcomes may involve combined effects that increase effectiveness, as well as conflicting interactions that could lessen the therapeutic benefits. Altered drug metabolism or distribution can lead to unforeseen side effects surfacing. The packaging of substances in EVs can alter their availability and uptake by cells, adding complexity to the situation [167]. Furthermore, natural substances could impact the composition or behavior of EVs, ultimately influencing their ability to transport substances. The diversity in EV makeup and the wide range of PCs create levels of intricacy [168]. Extensive preclinical research is essential for understanding these interactions, optimizing dosages, and determining the most optimal and safe combinations. Clinical trials need to thoroughly assess the effectiveness and safety characteristics of these intricate therapeutic systems [169]. This complex interaction highlights the importance of a comprehensive, methodical strategy in advancing these potentially difficult combination treatments.

Regulatory Hurdles: The use of natural compounds and EVs together in treatment faces intricate regulatory scrutiny because of their combination of multiple components. Regulatory agencies demand thorough proof of safety, effectiveness, and quality for every individual component as well as their combined form [170]. This requires thorough preclinical and clinical research, which is frequently more extensive than that needed for single-drug treatments. Regulators face obstacles in standardization and quality control due to the inherent variability in PC composition and EV characteristics [171]. Differences in international regulations add to the complexity of obtaining global approvals, as standards can differ greatly across regions. Demonstrating the additional advantage of the combination in comparison to current treatments introduces more complexity. All of these variables together lead to a longer, more expensive, and resource-heavy approval process, which could delay the implementation of these groundbreaking treatments from laboratory to patient care [172].

Manufacturing and Scalability: Developing and scaling EV and PC formulations for clinical use is technically challenging. Consistency in isolating EVs, purifying them, and loading cargo is crucial [173]. Creating large cell culture systems is vital for maintaining cellular health and EV quality. Standardized extraction, purification, and formulation processes are necessary due to variations in plant sources [174]. Combining EV production with natural compound manufacturing complicates matters. Analytical techniques are essential for assessing product quality. Stable storage and transportation are key [175]. Manufacturing complexity impacts cost, timelines, and feasibility of therapies, necessitating careful production design and validation for scalability without sacrificing effectiveness [176].

## 17. Future Perspective

In the future, research should focus on key areas to improve the effectiveness of combination therapies with 5-FU, natural compounds, and EVs. First and foremost, it is crucial to optimize methods of delivering EVs in order to improve strategies for targeted and efficient drug delivery. This entails investigating different ways of administration, increasing the stability of EVs, and improving their targeting of cancer cells. Additionally, it is important to keep working on finding natural compounds that work well with 5-FU to improve its therapeutic benefits. Evaluating various natural products for their capacity to influence pathways like p53 and other important mechanisms may reveal new combinations with potential for enhanced therapeutic effects. Finally, it is crucial to perform thorough mechanistic investigations in order to understand the complex relationships among PCs, EVs, 5-FU, and cellular pathways such as p53. These research projects need to concentrate on understanding molecular interactions, signaling pathways, and genetic elements that impact synergistic outcomes, offering valuable perspectives for enhancing combination treatments and combating drug resistance in cancer therapy.

## 18. Conclusions

In summary, the potential of combining 5-FU with PCs such as curcumin, resveratrol, and EGCG to target different aspects of cancer and overcome resistance mechanisms is promising. Nevertheless, validating these encouraging results for clinical application requires thorough verification via preclinical research and clinical testing. Validating these combinations, standardizing, and personalizing treatments according to patient profiles is essential. Additionally, researching EV-related biomolecules may reveal new diagnostic and prognostic markers for detecting and categorizing patients with CRC early on. Overcoming obstacles like tumor heterogeneity and refining EV isolation techniques are crucial for progressing EV-based therapeutic approaches. Combining immunomodulatory drugs with 5-FU and studying non-coding RNAs in EVs can improve treatment effectiveness and address resistance mechanisms. Using precision medicine tactics tailored to specific tumor traits and utilizing modified EVs for precise drug administration show potential for creating individualized methods in treating CRC. Immunotherapy strategies leveraging EV-mediated immune modulation and multi-omics approaches provide comprehensive insights for targeted interventions. Clinical validation through rigorous trials is imperative for realizing the full potential of these approaches in CRC management. Figure 5 demonstrates overcoming 5-FU resistance in CRC through exploring PCs and engineered EVs.

## Figures and Tables

**Figure 1 ijms-25-07470-f001:**
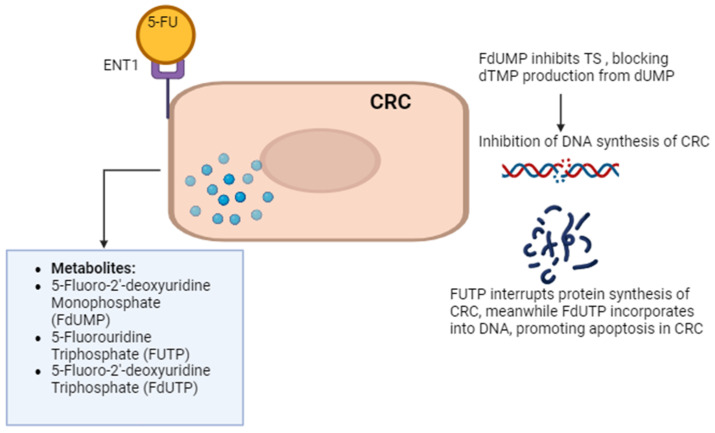
5-FU mechanism of action. 5-FU enters CRC cells via passive diffusion and active transport using transporters like equilibrative nucleoside transporter 1 (ENT1). Once inside, it is converted to active metabolites. 5-FU’s metabolites impact RNA and DNA functions, contributing to cell death. One such metabolite, 5-fluorodeoxyuridine monophosphate (FdUMP), inhibits thymidylate synthase (TS), disrupting DNA synthesis by blocking the formation of thymidine monophosphate (dTMP). Other 5-FU metabolites, such as 5-fluorouridine triphosphate (FUTP) and 5-fluoro-2’-deoxyuridine triphosphate (FdUTP), are incorporated into RNA and DNA, respectively, further disrupting their functions and contributing to cell death. The figure was created with BioRender.com.

**Figure 2 ijms-25-07470-f002:**
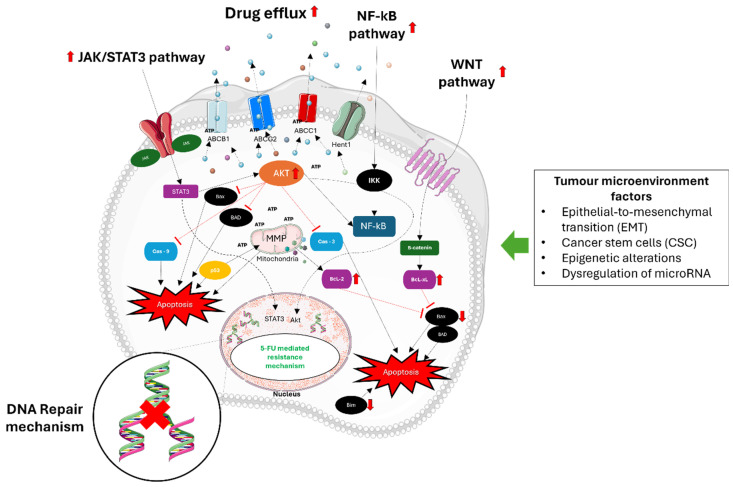
Mechanisms of 5-FU resistance in CRC. 5-FU resistance in CRC involves multiple complex mechanisms. Primary mechanisms involve improved DNA repair pathways like mismatch repair (MMR) and base excision repair (BER), which assist in fixing DNA damage caused by 5-FU, ultimately decreasing its toxic impacts. Epithelial–mesenchymal transition (EMT) helps cancer cells gain the ability to migrate and invade, a key factor in developing resistance. Cancer stem cells (CSCs) have the ability to self-renew and differentiate, making them naturally resistant to chemotherapy drugs like 5-FU. MiRNAs that are not functioning properly alter the expression of genes involved in drug sensitivity and resistance pathways. DNA methylation and histone modifications can silence tumor suppressor genes and activate genes associated with resistance. Moreover, important signaling pathways such as Wnt/β-catenin, NF-κB, JAK/STAT, and Akt are frequently activated in CRCs that are resistant to 5-FU. Stimulation of these pathways enhances cell growth, viability, and the ability to resist cell death, thus aiding in chemoresistance. Moreover, the upregulation of ATP-binding cassette (ABC) transporters enables a higher efflux of 5-FU from cancer cells, leading to decreased intracellular levels and effectiveness of the treatment. Akt, also referred to as protein kinase B (PKB), has a key role in colorectal cancer (CRC) resistance to 5-FU chemotherapy. Activation of Akt prevents cell death by blocking apoptosis, a vital response triggered by chemotherapy agents such as 5-FU. This hindrance allows cancer cells to escape the toxic effects of 5-FU, thus playing a major role in treatment resistance. Problems in Akt signaling in CRC can happen in different ways, like genetic mutations, amplifications, or changes in growth factors and receptor tyrosine kinases that control it. Moreover, the process of epithelial–mesenchymal transition (EMT) plays a pivotal role in promoting metastasis and resistance to chemotherapy in CRC. EMT involves the transformation of epithelial cells into mesenchymal-like cells, enhancing cancer cell migration, invasion, and resistance to apoptosis-inducing therapies like 5-FU. Understanding these mechanisms is crucial for developing new strategies to overcome 5-FU resistance in CRC. The figure was created with Microsoft PowerPoint, version 2406, accessed on 1 July 2024.

**Figure 3 ijms-25-07470-f003:**
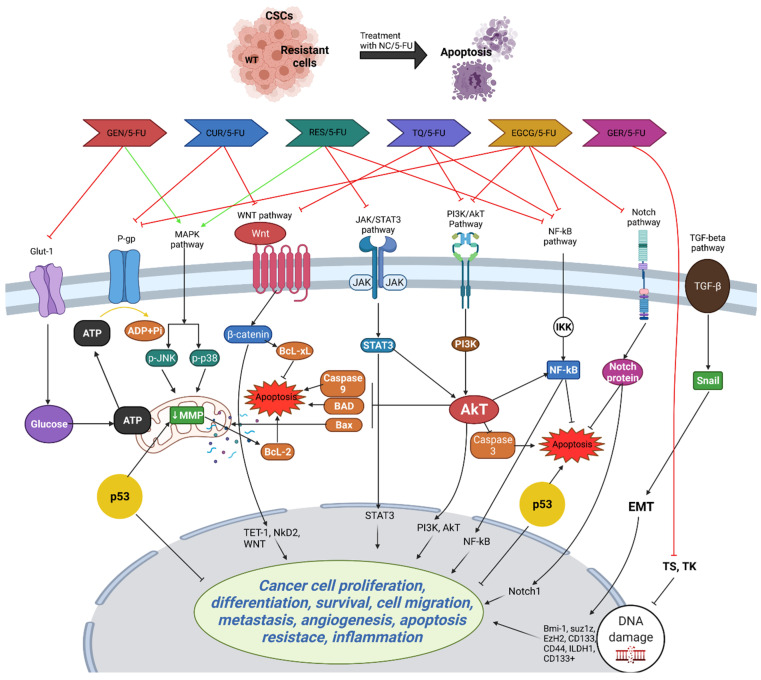
Potential action of PCs that can be used in combination with 5-FU. Combining 5-FU with PCs like curcumin (CUR), resveratrol (RES), epigallocatechin gallate (EGCG), genistein (GEN), geraniol (GER), and thymoquinone (TQ) enhances its efficacy in treating CRC by targeting various resistance mechanisms. Curcumin induces apoptosis, inhibits COX-2, and suppresses cancer stem cells (CSCs) and epithelial–mesenchymal transition (EMT), while resveratrol triggers apoptosis, inhibits cell cycle arrest, and suppresses the NF-kB pathway. EGCG enhances cytotoxicity and targets CSCs, genistein promotes apoptosis and inhibits COX-2 production, geraniol increases 5-FU uptake and reduces TS levels, and TQ targets CSCs and inhibits key signaling pathways like Wnt/β-catenin and PI3K/Akt. These combinations collectively improve 5-FU’s cytotoxic effects, making it more effective against chemoresistant CRC cells. The figure was created with Microsoft PowerPoint.

**Figure 4 ijms-25-07470-f004:**
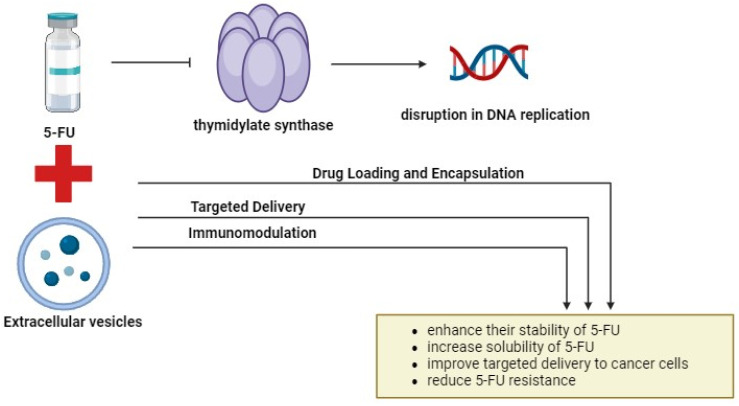
The combination of 5-FU and EVs. Combining 5-FU with EVs offers a promising approach to cancer treatment. By encapsulating 5-FU into EVs, their stability, solubility, and targeted delivery to cancer cells can be enhanced. Engineered EVs with specific surface markers enable precise cancer cell targeting, minimizing off-target effects, and improving therapeutic efficacy. Additionally, EV-mediated delivery of immunomodulatory agents enhances antitumor immunity and targets cancer stem cells. This innovative approach holds significant promise for improving cancer treatment outcomes by leveraging the unique capabilities of EVs. The figure was created with BioRender.com.

**Figure 5 ijms-25-07470-f005:**
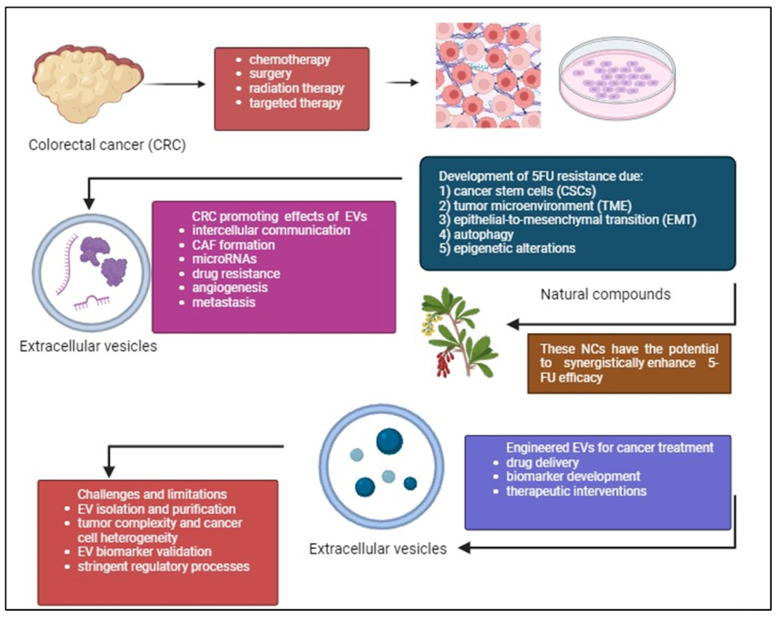
CRC treatment often faces the challenge of 5-FU drug resistance despite existing treatments such as chemotherapy, radiation therapy, surgery, targeted therapy. CSCs, complexity of TME, EMT, autophagy, and epigenetic alterations are the causes of this resistance. Phytochemicals (PCs) can work together with 5-FU to improve treatment effectiveness, reduce side effects, and reshape the tumor environment to make treatment more successful. Additionally, the potential of using engineered EVs to deliver drugs, target cancer cells, and serve as biomarkers is discussed. While challenges remain, this area of research is promising for future CRC treatments. The figure was created with BioRender.com.

**Table 1 ijms-25-07470-t001:** Phytochemicals and their effects on 5-FU-resistant CRC.

Natural Compound	Description	Mechanism of Action	Combination with 5-FU
Curcumin	Polyphenol extracted from turmeric	Induces apoptosis, inhibits COX-2, suppresses CSCs and EMT, downregulates Nrfs, inhibits WNT pathway	Enhances cytotoxicity, reduces COX-2 protein, inhibits CSCs and EMT
Resveratrol	Stilbenoid found in grapes, blueberries, etc.	Induces apoptosis, downregulates cyclin D1, E2 and BCL-2, upregulates BAX and p53	Induces apoptosis via caspase-6 activation, inhibits cell cycle arrest at S phase, sensitizes cells to oxidative stress, inhibits EMT, suppresses NF-kB pathway
Epigallocatechin Gallate (EGCG)	Most potent antioxidant in green tea	Induces apoptosis through caspase-3/7 and PERK activation	Enhances cytotoxicity, downregulates MDR1 expression, targets cancer stem cells
Genistein	Isoflavone found in soy products	Inhibits cell migration, reverses EMT, affects Notch-1, NF-kB, and PI3K/Akt pathways	Promotes apoptosis, downregulates Glut-1, upregulates p21, inhibits COX-2 production
Geraniol	Monoterpenic alcohol found in essential oils	Induces apoptosis, enhances 5-FU efficacy	Increases 5-FU intracellular uptake, reduces TS and TK levels
Thymoquinone (TQ)	Bioactive compound found in black cumin	Induces apoptosis, inhibits STAT3 signaling, suppresses CSC markers	Enhances cytotoxicity, targets CSCs, downregulates Wnt/β-catenin and PI3K/Akt pathways

Note: 5-FU = 5-fluorouracil; BAX = bcl-2-like protein 4; BCL-2 = B-cell lymphoma 2; COX-2 = cyclooxygenase-2; CSC = cancer stem cells; EMT = epithelial–mesenchymal transition; Glut-1 = Glucose transporter 1; MDR1 = multiple drug resistance 1; NF-kB = nuclear factor kappa-light-chain-enhancer of activated B; Nrfs = nuclear respiratory factors; PERK = protein kinase R-like ER kinase; PI3K/Akt = phosphatidylinositol 3-kinase; STAT3 = signal transducer and activator of transcription 3; TK = thymidine kinase; TS = thymidylate synthase.

**Table 2 ijms-25-07470-t002:** CRC-promoting effects of EVs.

CRC-Promoting Effect of EVs	Evidence	References
Induce formation of CAFs	Reprogramming proteome of CAFs by inducing TGF-β leading to the establishment of a favorable TME for cancer growth	[114]
	Circ_0067557, highly expressed in CAF-derived exosomes and promotes CRC progression and chemoresistance by targeting Lin28A and Lin28B	[116]
	CAFs from CRC exosomes promote EMT through downregulation of E-cadherin and upregulation of N-cadherin and β-catenin contributing to premetastatic niche formation	[117]
	CAFs-derived exosomes containing miR-24-3p promote tumor growth and resistance to methotrexate (MTX) by targeting the CDX2/HPEH regulatory axis	[118]
	CAFs releasing exosomes containing miR-17-5p prompted CAFs to CRC cells, and regulation of cell behavior leading to tumor progression	[119]
	CRC exosomal miR-10b promotes CAF formation by suppressing PIK3CA gene, activating TGF-β and α-SMA production	[127]
Promote angiogenesis	Exosomes derived from hypoxic CRC cells induce Wnt4-induced β-catenin signaling related to angiogenesis in endothelial cells	[120]
	B7-H3 in exosomes activates the AKT1/mTOR/VEGFA signaling pathway in VECs responsible for migration, invasion, and angiogenesis in CRC	[121]
	CAF-derived exosomal miR-135b-5p promotes CRC growth and angiogenesis by inhibiting TXNIP	[124]
Promote metastasis	Exosomal PAD4 affects the citrullination of the extracellular matrix (ECM), promoting colon cancer EMT and liver metastasis	[122]
	Exosomal HSPC111 promotes CRC liver metastasis by upregulating acetyl-CoA levels and tumor-promoting proteins in CAFs	[123]
	CAF-derived exosomal miR-17-5 promotes metastasis by suppressing RUNX3 in CRC	[119]
	Exosomal circEIF3K promotes malignant growth, metastasis by regulating the miR-214/PD-L1 axis	[130]
	Exosomes derived from chemoresistant HCT-15/FU cells promoted angiogenesis through GDF15 and inhibition of Smad signaling	[132]
	Exosomal miR-1246/92b-3p/27a-3p, CXCL16/RhoA/IL-8 (Fusobacterium nucleatum) promotes metastasis	[137]
Induce drug resistance	Exosomal miR-21 promotes proliferation, invasion, and therapy resistance by suppressing PDCD4, TPM1, PTEN	[125]
	CAF-derived exosomal miR-625-3p promotes invasion, migration, and chemoresistance	[131]
	CAF-derived exosomal miR-181d-5p inhibits 5-FU sensitivity via the METTL3/miR-181d-5p/NCALD axis	[133]
	Exosomal Nrf2 (oxaliplatin-resistant CRC cells) promotes oxaliplatin resistance by increasing Nrf2 levels in recipient cells	[134]
	Exosomal miR-92a-3p (stromal cells) promotes EMT, and inhibits apoptosis, metastasis, and chemoresistance by activating the Wnt/β-catenin pathway, suppressing FBXW7 and MOAP1	[135]
	Exosomal p-STAT3 promotes chemoresistance	[136]
Promote tumor proliferation	Exosomal miR-2149-5p, miR-6737-5p, and miR-6819-5 promote tumor proliferation by inhibiting TP53 expression in fibroblasts	[126]
	Exosomal miR-181b-3p promotes proliferation and migration, and decreases apoptosis by targeting SNX2	[129]
Induce EMT and inhibit apoptosis	Exosomal miR-224-5p promotes proliferation, migration, and invasion, and inhibits apoptosis by targeting SLC4A4 expression	[128]
	Exosomal CricN4BP2L2 promotes oxaliplatin resistance and stemness, and inhibits apoptosis by regulating the PI3K/AKT/mTOR pathway	[133]

**Table 3 ijms-25-07470-t003:** Potential applications of EVs in CRC.

Type of EVs/Compounds	Effects on CRC	References
Engineered EVs	Inhibition of CRC cell proliferation, migration, and invasion both in vitro and in vivo	[142,143,144,145,146]
EV-associated biomolecules (miRNAs, proteins)	Differentiation of CRC from healthy tissue, prediction of patient outcomes	[147,148]
Plant-derived exosome-like nanoparticles (PENs)	Delivery of therapeutic agents, including CRISPR/Cas9 components targeting lncRNAs associated with CRC	[149,150,151]
Grape-derived exosome-like nanoparticles (PDENs)	Specific treatment of colon cancer, improved therapeutic outcomes	[152,153,154]
Exosomes derived from CT26 tumor cells	Reduction in tumor size, enhanced immune response	[154]
Grapefruit-derived extracellular vesicles (GfEVs)	Safe and effective delivery of proteins into peripheral blood mononuclear cells and colon cancer cells	[155,156]
Lemon-derived exosome-like nanoparticles	Potential therapeutic intervention	[157]
Clinical trials with plant-derived exosomes	Evaluation of potential therapeutic effects in colon cancer patients	[158,159]
Exosomes derived from BMSCs and ascites	Inhibition of CRC cell proliferation, migration, and invasion, induction of apoptosis	[160,161]
ANGPTL1-containing exosomes	Attenuation of CRC liver metastasis	[162]
Exosomal DNA of CRC cells	Stimulation of antitumor immunity, activation of dendritic cells, and subsequent CD8+ T cell activation	[163,164]
Exosomes from non-CRC origins (e.g., BMSCs)	Inhibition of CRC cell proliferation and invasion	[165]

## Data Availability

Not applicable.

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
