# Peer review of "Innovative Strategies to Combat 5-Fluorouracil Resistance in Colorectal Cancer: The Role of Phytochemicals and Extracellular Vesicles"

_ijms, 2024, doi:10.3390/ijms25137470_

Round 1

Reviewer 1 Report

Comments and Suggestions for Authors

The present review article (A Next-Gen Colorectal Cancer Therapy: Combining 5-FU with 2 Naturals and EVs) has a promising sound in the clinical field, which summarizes the recent strategies to increase the therapeutic activity of 5-FU during the treatment of colorectal cancer. In the current review, the authors focus on using green approaches to increase the therapeutic activity of 5-FU and reduce drug resistance using two approaches. Firstly, the concurrent administration of natural compounds with 5-FU significantly enhances its cytotoxic activity with minimal toxicity. Secondly, using extracellular vesicles (EVs) as drug carriers to increase the internalization of drugs within cancer cells. However, below is a list of information that could help the authors to improve the manuscript.

1.     Title

Please do not use abbreviations in the title (5-FU)

In the present review, two green approaches have been summarized.

The first one is the adjunctive administration of natural compounds. Therefore, they are green in terms of toxicity owing to their presence in nature, where they are ingested in food. However, please try to use more significant terms like Nutraceutical and Phytochemicals.

The second is extracellular vesicles, which are normally present in the human body as carriers for intrinsic cell components. Therefore, they are green compared to other synthetic drug delivery systems.

Therefore, I suggest changing the title. Find below four suggested titles from which you can make the final title.

Ø  Green Approaches to Overcome 5-Fluorouracil Resistance in Colorectal Cancer Treatment: Nutraceutical Compounds and Extracellular Vesicles

Ø  Innovative Strategies to Combat 5-Fluorouracil Resistance in Colorectal Cancer: The Role of Phytochemicals Compounds and Extracellular Vesicles

Ø  Green Innovations in Colorectal Cancer Therapy: Nutraceuticals and Extracellular Vesicles to Overcome 5-Fluorouracil Resistance

Ø  Green Synergies in Colorectal Cancer Therapy: Phytochemicals Compounds and Extracellular Vesicles Approaches to Overcome 5-Fluorouracil Resistance

2.     Abstract

Please try to restructure the abstract using the advantages of your abovementioned approaches as two solutions for 5- Fluorouracil Resistance.

3.     Colorectal can cancer

Please mention some statistics and risk factors for colorectal cancer. This could add benefit for the readers by avoiding exposure to these conditions.

https://www.who.int/news-room/fact-sheets/detail/colorectal-cancer?gad_source=1&gclid=Cj0KCQjwsuSzBhCLARIsAIcdLm6ew117aX3mm5235GOuXNQrZjt5oz5o4S8QGyiVXInVubBWIPCzrPAaAh32EALw_wcB

4.     Combination of Natural Compound and 5-FU Strategy

Please change this title to (adjunctive Nutraceutical or Phytochemicals).

In addition, add 6 to 11 titles as subtitles to section 4. For example, change (6 NCs: Curcumin) to (Curcumin) and remove NCs from all other titles.

5.     General comments

·       Try to revise the abbreviations and mention them only the first time. For example, colorectal cancer in line 66. Use the search tool in Word to help you.

Author Response

Dear Reviewer 1,

Greetings!

We are pleased to submit our revised manuscript titled “Innovative Strategies to Combat 5-Fluorouracil Resistance in Colorectal Cancer: The Role of Phytochemicals Compounds and Extracellular Vesicles” for publication in your esteemed journal. We have highlighted the changes that was made in the manuscript according to the recommendation from the reviewer using point-by-point format.

The present review article (A Next-Gen Colorectal Cancer Therapy: Combining 5-FU with 2 Naturals and EVs) has a promising sound in the clinical field, which summarizes the recent strategies to increase the therapeutic activity of 5-FU during the treatment of colorectal cancer. In the current review, the authors focus on using green approaches to increase the therapeutic activity of 5-FU and reduce drug resistance using two approaches. Firstly, the concurrent administration of natural compounds with 5-FU significantly enhances its cytotoxic activity with minimal toxicity. Secondly, using extracellular vesicles (EVs) as drug carriers to increase the internalization of drugs within cancer cells. However, below is a list of information that could help the authors to improve the manuscript.

Reply: Thank you for your recommendation. We have now improved the manuscript to reviewer’s suggestion.

  1. Title: Please do not use abbreviations in the title (5-FU). In the present review, two green approaches have been summarized. The first one is the adjunctive administration of natural compounds. Therefore, they are green in terms of toxicity owing to their presence in nature, where they are ingested in food. However, please try to use more significant terms like Nutraceutical and Phytochemicals. The second is extracellular vesicles, which are normally present in the human body as carriers for intrinsic cell components. Therefore, they are green compared to other synthetic drug delivery systems. Therefore, I suggest changing the title. Find below four suggested titles from which you can make the final title.

Ø Green Approaches to Overcome 5-Fluorouracil Resistance in Colorectal Cancer Treatment: Nutraceutical Compounds and Extracellular Vesicles

Ø Innovative Strategies to Combat 5-Fluorouracil Resistance in Colorectal Cancer: The Role of Phytochemicals Compounds and Extracellular Vesicles

Ø Green Innovations in Colorectal Cancer Therapy: Nutraceuticals and Extracellular Vesicles to Overcome 5-Fluorouracil Resistance

Ø Green Synergies in Colorectal Cancer Therapy: Phytochemicals Compounds and Extracellular Vesicles Approaches to Overcome 5-Fluorouracil Resistance

Reply: Thank you for your recommendation. We have improved our manuscript accordingly and selected the new title: 'Innovative Strategies to Combat 5-Fluorouracil Resistance in Colorectal Cancer: The Role of Phytochemical Compounds and Extracellular Vesicles.'

  1. Abstract: Please try to restructure the abstract using the advantages of your abovementioned approaches as two solutions for 5- Fluorouracil Resistance.

Reply: Thank you for your suggestion. We have improved the abstract in the revised manuscript (line 24-28)

  1. Colorectal can cancer: Please mention some statistics and risk factors for colorectal cancer. This could add benefit for the readers by avoiding exposure to these conditions.

Reply: Thank you for your suggestion. We have included the risk factors in the revised manuscript (lines 56-62), and the statistics have already been added (lines 63-74).

  1. Combination of Natural Compound and 5-FU Strategy: Please change this title to (adjunctive Nutraceutical or Phytochemicals). In addition, add 6 to 11 titles as subtitles to section 4. For example, change (6 NCs: Curcumin) to (Curcumin) and remove NCs from all other titles.

Reply: Thank you for the suggestion. We have corrected it throughout the manuscript

  1. General comments: Try to revise the abbreviations and mention them only the first time. For example, colorectal cancer in line 66. Use the search tool in Word to help you.

Reply: Thank you for the suggestion. We have corrected it throughout the manuscript

Thank you for the opportunity to further explain our research hence we request you to consider the manuscript for publication in the esteemed journal and oblige.

Thank you.

Sincerely,

Barathan Muttiah

Reviewer 2 Report

Comments and Suggestions for Authors

The manuscript presents an outlook over the use new combinations of 5-Fu and natural products and new delivery systems for the treatment of colorectal cancer. The data stresses on the gaps in this regard. Comments:

1. Colorectal cancer: The causes and elicitors for CRC should be added and discussed.

2. Current treatments and application of 5-FU: data from different countries should be included indicating the response percentages for different treatment regimens and histograms will be of benefit in this part.

3. Application of 5-FU: the mechanism of action should be represented as well as a figure, that will be more informative.

4. Figure 1 (5-FU resistance) and 2 (Nc reversal of 5-FU resistance), maybe it is better to combine them in one figure.

5. Is it possible to speculate new combinations based on the collected data. Such as using more than one natural compound with 5-FU to reverse more than one site of resistance?. A bit of network pharmacology will add greatly in this regard.

6. The software used for creating figures must be mentioned and cited.

Author Response

Dear Reviewer 2,

Greetings!

We are pleased to submit our revised manuscript titled “Innovative Strategies to Combat 5-Fluorouracil Resistance in Colorectal Cancer: The Role of Phytochemicals Compounds and Extracellular Vesicles” for publication in your esteemed journal. We have highlighted the changes that was made in the manuscript according to the recommendation from the reviewer using point-by-point format.

The manuscript presents an outlook over the use new combinations of 5-Fu and natural products and new delivery systems for the treatment of colorectal cancer. The data stresses on the gaps in this regard.

Reply: Thank you for your recommendation. We have now improved the manuscript to reviewer’s suggestion.

  1. Colorectal cancer: The causes and elicitors for CRC should be added and discussed.

Reply: Thank you for your suggestion. We have included the risk factors in the revised manuscript (lines 56-62), and the statistics have already been added (lines 63-74).

  1. Current treatments and application of 5-FU: data from different countries should be included indicating the response percentages for different treatment regimens and histograms will be of benefit in this part.

Reply: Thank you for your suggestion. We greatly value your input. Our review article aims to delve deeply into the therapeutic potential of natural products and extracellular vesicles in the context of colon cancer treatment. While the inclusion of response percentages from various countries and the use of histograms could be beneficial for certain aspects of our research, our primary objective is to comprehensively analyze how these innovative therapies can impact colon cancer management. We apologize for any inconvenience this may cause and appreciate your understanding.

  1. Application of 5-FU: the mechanism of action should be represented as well as a figure, that will be more informative.

Reply: Thank you for your suggestion. We have now added a related figure in the revised manuscript. (line 104-113)

  1. Figure 1 (5-FU resistance) and 2 (Nc reversal of 5-FU resistance), maybe it is better to combine them in one figure.

Reply: Thank you for the suggestion. Combining Figure 1 (5-FU resistance) and Figure 2 (Nc reversal of 5-FU resistance) into a single figure may enhance their clarity and coherence. However, there is concern that this consolidation could potentially complicate the visual presentation of the data, leading to difficulty in interpreting the responses depicted. Therefore, to maintain clarity and ease of understanding, it may be preferable to retain these figures separately. We apologize for this.

  1. Is it possible to speculate new combinations based on the collected data. Such as using more than one natural compound with 5-FU to reverse more than one site of resistance?. A bit of network pharmacology will add greatly in this regard.

Reply: Thank you for your suggestion. There are many researchers on using one natural compound with 5-FU, we have reported some in the review article however combining more than two natural compounds with 5-FU may alter their pharmacokinetics and pharmacodynamics, leading to unexpected side effects or reduced efficacy. While many natural compounds are generally considered safe, high doses or specific combinations could cause toxicity, necessitating rigorous testing in preclinical and clinical settings. Additionally, natural compounds can vary in composition and potency, making it challenging to ensure consistent dosing and effects, thus standardization of extracts and compounds is crucial. By mapping the molecular targets and pathways involved, we can predict the combined effects and optimize treatment regimens. Network pharmacology can also help identify potential adverse interactions and guide the design of combination therapies.

  1. The software used for creating figures must be mentioned and cited.

Reply: Thank you for the suggestion. All of the figures were created using Microsoft PowerPoint and Biorender. They have been cited accordingly.

Thank you for the opportunity to further explain our research hence we request you to consider the manuscript for publication in the esteemed journal and oblige.

Thank you.

Sincerely,

Barathan Muttiah

Reviewer 3 Report

Comments and Suggestions for Authors

Barathan et al. conducted a review on combining 5-FU with two natural compounds and EVs for colon cancer treatment. It is an interesting topic.

Combining 5-Fluorouracil (5-FU) with natural compounds and extracellular vesicles (EVs) for colon cancer therapy offers a promising multifaceted approach.

Here are some key points to consider: Synergistic Effects: 1. 5-Fluorouracil (5-FU): - Mechanism: 5-FU is a chemotherapy drug that inhibits thymidylate synthase, disrupting DNA synthesis and inducing cell death in rapidly dividing cancer cells.  2. Natural Compounds: - Potential Benefits: 3. Extracellular Vesicles (EVs): - Mechanism: EVs can deliver bioactive molecules, such as miRNAs, proteins, and drugs, directly to cancer cells, enhancing therapeutic efficacy and reducing off-target effects. - Advantages: EVs offer targeted delivery, potentially improving the therapeutic index of 5-FU and natural compounds.

Authors should add one section to discuss Potential Challenges. These may include the following: 1. Complex Interactions: The interactions between different compounds and delivery systems can be complex and unpredictable, requiring extensive research. 2. Regulatory Hurdles: Regulatory approval processes for combination therapies can be more challenging compared to single-agent therapies. 3. Manufacturing and Scalability: Producing and standardizing EVs and natural compound formulations on a large scale can be technically challenging.

 Additionally, the authors should add a section about the role of p53 in mediating the cellular response to DNA damage induced by 5-FU. It is important to explain how p53 can induce apoptosis and cell cycle arrest, thereby enhancing the cytotoxic effects of 5-FU. This can provide valuable insights into the molecular mechanisms underlying the therapeutic effects. Furthermore, the authors can suggest areas for further research, such as optimizing the delivery methods for EVs, identifying additional natural compounds that can synergize with 5-FU and p53, and conducting detailed mechanistic studies.

Author Response

Dear Reviewer 3,

Greetings!

We are pleased to submit our revised manuscript titled “Innovative Strategies to Combat 5-Fluorouracil Resistance in Colorectal Cancer: The Role of Phytochemicals Compounds and Extracellular Vesicles” for publication in your esteemed journal. We have highlighted the changes that was made in the manuscript according to the recommendation from the reviewer using point-by-point format.

Barathan et al. conducted a review on combining 5-FU with two natural compounds and EVs for colon cancer treatment. It is an interesting topic. Combining 5-Fluorouracil (5-FU) with natural compounds and extracellular vesicles (EVs) for colon cancer therapy offers a promising multifaceted approach.

Reply: Thank you for your recommendation. We have now improved the manuscript to reviewer’s suggestion.

  1. Synergistic Effects: 1. 5-Fluorouracil (5-FU): - Mechanism: 5-FU is a chemotherapy drug that inhibits thymidylate synthase, disrupting DNA synthesis and inducing cell death in rapidly dividing cancer cells.

Reply: Thank you for your suggestion. We have now added the 5-FU mechanism of action. (line 104-113)

  1. Natural Compounds: - Potential Benefits.

Reply: Thank you for your suggestion. We have now added the Natural Compounds: - Potential Benefits. (line 221-220)

  1. Extracellular Vesicles (EVs): - Mechanism: EVs can deliver bioactive molecules, such as miRNAs, proteins, and drugs, directly to cancer cells, enhancing therapeutic efficacy and reducing off-target effects. - Advantages: EVs offer targeted delivery, potentially improving the therapeutic index of 5-FU and natural compounds.

Reply: Thank you for your suggestion. We have now added the extracellular vesicles (EVs): - mechanism and advantages. (line 494-502)

  1. Authors should add one section to discuss Potential Challenges. These may include the following: 1. Complex Interactions: The interactions between different compounds and delivery systems can be complex and unpredictable, requiring extensive research. 2. Regulatory Hurdles: Regulatory approval processes for combination therapies can be more challenging compared to single-agent therapies. 3. Manufacturing and Scalability: Producing and standardizing EVs and natural compound formulations on a large scale can be technically challenging.

Reply: Thank you for the suggestion. We have added relevant related information in this revised manuscript. (line 801-840)

  1. Additionally, the authors should add a section about the role of p53 in mediating the cellular response to DNA damage induced by 5-FU. It is important to explain how p53 can induce apoptosis and cell cycle arrest, thereby enhancing the cytotoxic effects of 5-FU. This can provide valuable insights into the molecular mechanisms underlying the therapeutic effects. Furthermore, the authors can suggest areas for further research, such as optimizing the delivery methods for EVs, identifying additional natural compounds that can synergize with 5-FU and p53, and conducting detailed mechanistic studies.

Reply: Thank you for the suggestion. We have added the relevant information to the revised manuscript. (line 149-156 (p53), line 853-867(future perspective)

Thank you for the opportunity to further explain our research hence we request you to consider the manuscript for publication in the esteemed journal and oblige.

Thank you.

Sincerely,

Barathan Muttiah

Reviewer 4 Report

Comments and Suggestions for Authors

The topic of the manuscript is quite interesting. However, there are several concerns which should be taken care of.

Abstract: The term „EVs“ should be explained.

Introduction: FOLFOX, CAPOX, and FOLFIRI should be explained, as well as the abbreviations VEGF, EGFR, DFS, MSI-H, dMMR, EBRT (chapter 2). Also chapter 3: TS. Please check whether all abbreviations have been explained when mentioned the first time.

Introduction: There are some repetitions concerning FU mechanism: E.g. chapter 2: „FU remains a cornerstone, severely damaging tumor cell RNA and DNA by inhibiting thymidylate synthase and serving as the primary choice in CRC chemotherapy. Chapter 3: „FU …severely damages the RNA and DNA … by inhibiting thymidylate synthase … as the primary choice in CRC chemotherapy“. Additionally in line 98. „considered the gold standard in the first cycle of CRC chemotherapy“. Line 101: „recognized as the superior choice in second-line treatment“. Line 102: „FU is taken up by cancer cells and converted into active forms that inhibit TS and integrate into RNA and DNA“. Line 107: „disrupting DNA synthesis“. Also line 208/209: „5-FU based regimens are still the most efficient drugs and the backbone for CRC chemotherapy“. This should be avoided.

Figure 1: This figure is not clearly explained. The figure legend points to autophagy, but only „apoptosis“ is highlighted in the figure. The figure also depicts MMP (matrix metalloproteinase?), Akt which both are not explained, neither in the manuscript text nor in the figure legend. On the other hand „thymidylate synthase“ is mentioned in the legend and manuscript text but not shown in the figure. „Cell cycle checkpoints“ (cyclins, cdks?) are listed in the text but not depicted in the figure. Overall, the manuscript text should be clearly related to the figure and vice versa. Significant improvement is required.

Chapter 5: The first para is difficult to understand, and 4-AAQB is not explained at all. It is also not clear why 4-AAQB as well as ginseng (second para) has been mentioned only here but not later. It is also not clear why the authors point to clopidogrel, olaparib, irinotecan, although the topic of the article has been related to FU, combined with natural (not chemically modified) compounds. The referee recommends to start this chapter with an explanation why natural compounds in particular may provide benefit in the treatment of cancer (instead of chemically designed drugs).

Chapter 5, lines 216-225: These informations have already been given before in chapter 4.

Chapter 5: As the authors noted correctly, there are numerous natural compounds available. Was there a particular reason why the authors concentrated on 6 compounds listed in the table?

Chapter 12+13: These chapters do not match with the title and rather deal with EVs in general as a new topic. Only very few information on „EV mediated therapy for CRC“ are given in chapter 14 (e.g. lines 651-670). However, in most cases plant-derived microvesicles have been used as drug delivery systems. Only references 158+159 deal with curcumin loaded nanoparticles. Still, both articles are review articles, not original ones. Notably, there is no link to FU treatment in all these chapters. Similar chapter 15, which does not provide any link to natural compounds.

Last figure: Should be figure 4.

Author Response

Dear Reviewer 4,

Greetings!

We are pleased to submit our revised manuscript titled “Innovative Strategies to Combat 5-Fluorouracil Resistance in Colorectal Cancer: The Role of Phytochemicals Compounds and Extracellular Vesicles” for publication in your esteemed journal. We have highlighted the changes that was made in the manuscript according to the recommendation from the reviewer using point-by-point format.

The topic of the manuscript is quite interesting. However, there are several concerns which should be taken care of.

Reply: Thank you for your recommendation. We have now improved the manuscript to reviewer’s suggestion.

  1. Abstract: The term „EVs“ should be explained.

Reply: Thank you for your suggestion. We have corrected it accordingly.

  1. Introduction: FOLFOX, CAPOX, and FOLFIRI should be explained, as well as the abbreviations VEGF, EGFR, DFS, MSI-H, dMMR, EBRT (chapter 2). Also chapter 3: TS. Please check whether all abbreviations have been explained when mentioned the first time.

Reply: Thank you for your suggestion. We have corrected them accordingly.

  1. Introduction: There are some repetitions concerning FU mechanism: E.g. chapter 2: „FU remains a cornerstone, severely damaging tumor cell RNA and DNA by inhibiting thymidylate synthase and serving as the primary choice in CRC chemotherapy. Chapter 3: „FU …severely damages the RNA and DNA … by inhibiting thymidylate synthase … as the primary choice in CRC chemotherapy“. Additionally in line 98. „considered the gold standard in the first cycle of CRC chemotherapy“. Line 101: „recognized as the superior choice in second-line treatment“. Line 102: „FU is taken up by cancer cells and converted into active forms that inhibit TS and integrate into RNA and DNA“. Line 107: „disrupting DNA synthesis“. Also line 208/209: „5-FU based regimens are still the most efficient drugs and the backbone for CRC chemotherapy“. This should be avoided.

Reply: Thank you for your suggestion. We have corrected them accordingly

  1. Figure 1: This figure is not clearly explained. The figure legend points to autophagy, but only „apoptosis“ is highlighted in the figure. The figure also depicts MMP (matrix metalloproteinase?), Akt which both are not explained, neither in the manuscript text nor in the figure legend. On the other hand „thymidylate synthase“ is mentioned in the legend and manuscript text but not shown in the figure. „Cell cycle checkpoints“ (cyclins, cdks?) are listed in the text but not depicted in the figure. Overall, the manuscript text should be clearly related to the figure and vice versa. Significant improvement is required.

Reply: Thank you for the suggestion. We have completely rewritten the figure legend in this revised manuscript.

  1. Chapter 5: The first para is difficult to understand, and 4-AAQB is not explained at all. It is also not clear why 4-AAQB as well as ginseng (second para) has been mentioned only here but not later. It is also not clear why the authors point to clopidogrel, olaparib, irinotecan, although the topic of the article has been related to FU, combined with natural (not chemically modified) compounds. The referee recommends to start this chapter with an explanation why natural compounds in particular may provide benefit in the treatment of cancer (instead of chemically designed drugs).

Reply: Thank you for the suggestion. We have removed the entire statement and added new statement of benefits of natural product against chemically designed drugs in this revised manuscript. (line 220-232).

  1. Chapter 5, lines 216-225: These informations have already been given before in chapter 4.

Reply: Thank you for the suggestion. We have removed the statement in this revised manuscript.

  1. Chapter 5: As the authors noted correctly, there are numerous natural compounds available. Was there a particular reason why the authors concentrated on 6 compounds listed in the table?

Reply: Thank you for the question. The selection of the six compounds listed in the table was driven by the diversity and documented bioactivity of natural products relevant to our research objectives. We aimed to encompass a variety of natural compounds known for their potential anticancer properties, ensuring a comprehensive exploration while maintaining feasibility in our study scope. Each chosen compound offers unique characteristics that align with our research goals, including their mechanisms of action, availability for experimental study, and potential for future therapeutic applications in cancer treatment. This focused approach allows us to systematically evaluate and compare these diverse natural products to identify promising candidates for further investigation. We acknowledge that there are many other potential natural products worth exploring, and this serves as a limitation of our study.

  1. Chapter 12+13: These chapters do not match with the title and rather deal with EVs in general as a new topic. Only very few information on „EV mediated therapy for CRC“ are given in chapter 14 (e.g. lines 651-670). However, in most cases plant-derived microvesicles have been used as drug delivery systems. Only references 158+159 deal with curcumin loaded nanoparticles. Still, both articles are review articles, not original ones. Notably, there is no link to FU treatment in all these chapters. Similar chapter 15, which does not provide any link to natural compounds.

Reply: Thank you for pointing it out. Chapter 12 discusses EVs as communication mediators in intercellular communication. It describes the diverse types of EVs, including exosomes, microvesicles, and apoptotic bodies, each with distinct biogenesis pathways and functions. It also highlights that EVs carry a cargo of biomolecules such as lipids, nucleic acids, carbohydrates, and proteins, which can be transferred between cells. This transfer of bioactive molecules facilitates communication and influences various biological processes, emphasizing the role of EVs as important mediators in cellular interactions and signaling.

Chapter 13 summarizes the pivotal role of EVs, particularly exosomes, in promoting various aspects of CRC progression and resistance. EVs facilitate intercellular communication by transporting bioactive molecules like proteins, RNA (including miRNAs), and DNA between cells. They contribute to key processes such as EMT, angiogenesis, immune modulation, and drug resistance through mechanisms involving signaling pathways like Wnt/β-catenin and STAT3. These findings underscore the potential of EVs as both biomarkers and therapeutic targets in managing CRC

Chapter 14 summarizes the EVs that offer several advantages, such as protecting cargo from degradation, enhancing bioavailability, and enabling targeted delivery to tumor sites. Moreover, EVs can be engineered to carry specific molecules, such as small RNAs or proteins, that sensitize CRC cells to 5-FU, potentially reversing drug resistance mechanisms. Preclinical studies have shown encouraging results, demonstrating the ability of EVs to overcome resistance and improve therapeutic outcomes in CRC models. Moving forward, optimizing EV-based delivery systems and validating their efficacy in clinical settings are crucial steps toward developing effective treatments for 5-FU-resistant CRC patients. Meanwhile reference no 158-159 are referring to clinical trials that these two review articles mentioned about it.

We believe that suggesting 5-FU treatment in these chapters would offer valuable guidance, particularly given the current limited research focus on this topic. Including such a section could consolidate existing knowledge, highlight research gaps requiring further exploration, and serve as a roadmap for future studies, thereby advancing the field of 5-FU treatment in the context of our research. Also in Chapter 15, we added some information relevant to phytochemicals in the section.

  1. Last figure: Should be figure 4.

Reply: Thank you for pointing it out. We have corrected it accordingly

Thank you for the opportunity to further explain our research hence we request you to consider the manuscript for publication in the esteemed journal and oblige.

Thank you.

Sincerely,

Barathan Muttiah

Round 2

Reviewer 1 Report

Comments and Suggestions for Authors

The author has addressed all the comments. Now, the manuscript can be accepted. 

Author Response

Dear Reviewer 1,

Greetings!

The author has addressed all the comments. Now, the manuscript can be accepted.

Reply: Thank you for your suggestion. It helped us improve our manuscript and make it more readable.

Thank you for the opportunity to further explain our research hence we request you to consider the manuscript for publication in the esteemed journal and oblige.

Thank you.

Sincerely,

Barathan Muttiah

Reviewer 2 Report

Comments and Suggestions for Authors

Dear authors, many thanks for your reply, clarification and modifications.

Author Response

Dear Reviewer 2,

Greetings!

Dear authors, many thanks for your reply, clarification and modifications.

Reply: Thank you for your suggestion. It helped us improve our manuscript and make it more readable.

Thank you for the opportunity to further explain our research hence we request you to consider the manuscript for publication in the esteemed journal and oblige.

Thank you.

Sincerely,

Barathan Muttiah

Reviewer 3 Report

Comments and Suggestions for Authors

The current version of Barathan et al. is much improved.  The authors have largely addressed the comments/questions raised with new data and the discussion, which made this study stronger and better suited for publication.  

Author Response

Dear Reviewer 3,

Greetings!

The current version of Barathan et al. is much improved.  The authors have largely addressed the comments/questions raised with new data and the discussion, which made this study stronger and better suited for publication. 

Reply: Thank you for your suggestion. It helped us improve our manuscript and make it more readable.

Thank you for the opportunity to further explain our research hence we request you to consider the manuscript for publication in the esteemed journal and oblige.

Thank you.

Sincerely,

Barathan Muttiah

Reviewer 4 Report

Comments and Suggestions for Authors

The authors have improved their manuscript. However, few comments are remaing which should be taken care of.

1. Obviously, use of phytochemicals or extracellular vesicles are two (partially) independent strategies to overcome FU resistance. This should be stated out more clearly in the title and chapter 12. In fact, there is still no clear link between EV and their use as transporters of natural compounds.

2. Legend of figure 2 requires further improvement. Particularly Akt signaling should be mentioned, since it seems to be a central part in the figure. Given that EMT drives resistance, then the term EMT should be included in the figure. 

Author Response

Dear Reviewer 4,

Greetings!

We are pleased to submit our revised manuscript titled “Innovative Strategies to Combat 5-Fluorouracil Resistance in Colorectal Cancer: The Role of Phytochemicals and Extracellular Vesicles” for publication in your esteemed journal. We have highlighted the changes that was made in the manuscript according to the recommendation from the reviewer using point-by-point format.

The authors have improved their manuscript. However, few comments are remaing which should be taken care of.

Reply: Thank you for your recommendation. We have now improved the manuscript to reviewer’s suggestion.

  1. Obviously, use of phytochemicals or extracellular vesicles are two (partially) independent strategies to overcome FU resistance. This should be stated out more clearly in the title and chapter 12. In fact, there is still no clear link between EV and their use as transporters of natural compounds.

Reply: Thank you for your recommendation. We have now improved the title of manuscript to reviewer’s suggestion.

"Innovative Strategies to Combat 5-Fluorouracil Resistance in Colorectal Cancer: The Roles of Phytochemicals and Extracellular Vesicles"

While we recognize that there is currently no definitive link between EVs and their use as transporters of PCs to overcome 5-fluorouracil resistance in colorectal cancer, our manuscript aims to explore the potential of this combination. The potential of this combination has been discussed in more detail in the revised manuscript.

  1. Legend of figure 2 requires further improvement. Particularly Akt signaling should be mentioned, since it seems to be a central part in the figure. Given that EMT drives resistance, then the term EMT should be included in the figure.

Reply: Thank you for your recommendation. We have now improved the figure legend in this revised manuscript.

Thank you for the opportunity to further explain our research hence we request you to consider the manuscript for publication in the esteemed journal and oblige.

Thank you.

Barathan Muttiah
